# LOWER-LEVEL DUALITY BASED PENALTY METHODS FOR HYPERPARAMETER OPTIMIZATION

## ABSTRACT

Hyperparameter optimization (HO) is essential in machine learning and can be structured as a bilevel optimization. However, many existing algorithms designed for addressing nonsmooth lower-level problems involve solving sequential sub-problems with high complexity. To tackle this challenge, we introduce penalty methods for solving HO based on strong duality between the lower level problem and its dual. We illustrate that the penalized problem closely approximates the optimal solutions of the original HO under certain conditions. In many real applications, the penalized problem is a weakly-convex objective with proximal-friendly constraints. Furthermore, we develop two fully first-order algorithms to solve the penalized problems. Theoretically, we prove the convergence of the proposed algorithms. We demonstrate the efficiency and superiority of our method across numerical experiments.

## 1 INTRODUCTION

In machine learning, the introduction of regularization terms is a common practice aimed at enhancing model generalization and controlling model complexity. This overarching framework can be articulated as an objective function that strikes a balance between data fitting and model simplicity:

$$\min_{\mathbf{x}} \ l(\mathbf{x}) + \sum_{i=1}^{r} \lambda_i R_i(\mathbf{x}). \tag{1}$$

In this formulation, $l(\mathbf{x})$ represents the loss function and $\boldsymbol{\lambda} = (\lambda_1, \lambda_2, ..., \lambda_r)$ encompasses hyperparameters, which are not derived from the learning algorithm but rather specified as inputs. Meanwhile, $R_i(\mathbf{x}), i = 1, 2, ..., r$ denotes the regularizers, which are considered in the form of norms in this paper, i.e. $R_i(\cdot) = \| \cdot \|$. The pursuit of optimal hyperparameters that enhance predictive performance is a vital task in machine learning, commonly referred to as hyperparameter optimization (Feurer & Hutter, 2019; Gao et al., 2022; Ye et al., 2021; 2023; Chen et al., 2024). In supervised learning, this process involves partitioning the dataset into training, validation, and test sets, solving (1) for various $\boldsymbol{\lambda}$ values, and selecting the best $(\boldsymbol{\lambda}, \mathbf{x}_{\boldsymbol{\lambda}})$ based on validation and training error. The quality of the selected hyperparameters is ultimately evaluated through the test error function. This structured approach can be encapsulated within a bilevel optimization framework (Dempe & Zemkoho, 2020):

$$\begin{aligned} \min_{\mathbf{x}_{\boldsymbol{\lambda}}, \boldsymbol{\lambda}} \quad & L(\mathbf{x}_{\boldsymbol{\lambda}}) \\ \text{s.t.} \quad & \mathbf{x}_{\boldsymbol{\lambda}} \in \arg\min_{\mathbf{x}} \left\{ l(\mathbf{x}) + \sum_{i=1}^{r} \lambda_i R_i(\mathbf{x}) \right\}. \end{aligned} \tag{2}$$

In this formulation, $L$ serves as the loss function on the validation set, defining the upper-level (UL) problem, while $l$ represents the training set loss function, constituting the lower-level (LL) problem alongside the regularization terms. The hyperparameters $\boldsymbol{\lambda}$ help delineate the trade-off between fitting the data and maintaining simplicity.

### 1.1 MAIN CONTRIBUTIONS

We summarize our main contributions as follows. We propose a penalty method based on lower-level duality for hyperparameter optimization (2), which is in the form of bilevel optimization with non-smooth lower-level problem. Our method avoids any implicit value functions and high-complexity

subproblems. Additionally, we introduce first-order algorithms to solve the penalization problem and provide theoretical proof of its convergence. Through experimental results, we demonstrate the superiority of our algorithm, highlighting its independence from any convex optimization solvers while showcasing its exceptional efficiency.

### 1.2 RELATED WORK

**Hyperparameters Optimization.** The existing literature presents various strategies for hyperparameter selection. Among the simplest model-free techniques are grid search (Injadat et al., 2020) and random search (Bergstra & Bengio, 2012). Additionally, Bayesian optimization (Bergstra et al., 2011; Snoek et al., 2012) serves as a sequential algorithm that selects future evaluation points by leveraging insights from prior outcomes. However, these gradient-free methods face significant challenges when dealing with a high number of parameters. To address this limitation, Feng & Simon (2018) introduces gradient-based techniques for hyperparameter tuning.

**Bilevel Optimization.** In general, the problem presented in (2) aligns with the format known as bilevel optimization (BLO), which is pertinent to a diverse array of data-driven challenges, including hyperparameter optimization (Maclaurin et al., 2015; Franceschi et al., 2018), meta-learning (Finn et al., 2017), and reinforcement learning (Shen et al., 2024; Stadie et al., 2020).

The initial strategies for addressing bilevel optimization problems primarily centered on gradient-based algorithms, which can be broadly classified into two categories based on their methods for computing hypergradients. Iterative Differentiation (ITD) involves unrolling the lower-level problem into gradient steps and subsequently utilizing backpropagation to calculate the hypergradient (Franceschi et al., 2017; 2018; Grazzi et al., 2020; Liu et al., 2021b; Antoniou et al., 2018; Shaban et al., 2019). In contrast, Implicit Differentiation (AID) leverages the first-order optimality conditions of the lower-level problem along with the implicit function theorem to derive the hypergradient (Pedregosa, 2016; Rajeswaran et al., 2019; Lorraine et al., 2020; Yang et al., 2021; 2023). However, these methods necessitate the strong convexity of the lower-level problem, thereby constraining their applicability.

Recently, Chen et al. (2023a); Li et al. (2022); Chen et al. (2023b) have introduced a series of fully first-order methods that operate without requiring Hessian computations or implicit gradients. Additionally, many machine learning problems may exhibit multiple minima for the lower-level function. To address this challenge, Liu et al. (2021a) propose a value function based on the optimal value of the lower-level function, which leads to the development of novel algorithms employing a penalization technique (Liu et al., 2023). As a result, penalty-based methods have also emerged as effective solutions for bilevel optimization problems. Shen & Chen (2023); Lu & Mei (2024); Kwon et al. (2023b;a); Liu et al. (2022) construct single-level reformulation for original BLO by penalty method with various penalty terms.

**Nonsmooth Lower-level Problem.** When the regularazer is $l_1$ norm, Bertrand et al. (2020) proposes an implicit differentiation method with block coordinate descent for Lasso-type hyperparameter optimization, later extended to general nonsmooth problems Bertrand et al. (2022). Ye et al. (2021; 2023) utilize diffenrence-of-convex (DC) method for hyperparameter selection, while Gao et al. (2022) combine penalization with DC method for bilevel problems with nonsmooth regularizer. Both methods require computing the lower-level optimal value for subgradients. Recently, Chen et al. (2023a) propose an inexact gradient-free method, though the subproblem remains difficult to solve. Chen et al. (2024) presents a novel reformulation based on LL duality with no value function involved and proposes an iterative algorithm grounded in cone programming for many pratical applications alongside its corresponding off-the-shelf solver. Recent studies have also employed the Moreau envelope to effectively address nonsmooth functions. Works by Gao et al. (2023); Yao et al. (2024b); Liu et al. (2024) have restructured the original bilevel optimization framework using this strategy and propose a series of Moreau envelope-based algorithms, which demonstrate the capability to identify well-defined KKT points.

## 2 PENALIZATION FRAMEWORK

In this section, we introduce our **l**ower-level **d**uality based **p**enalty **m**ethod (LDPM) for hyperparameter optimization (2). We begin by separating and simplifying the hierarchical structure of the

lower-level problem using Fenchel duality. Unlike traditional primal-dual methods, we employ conjugate functions to transform the subproblems into constrained optimization problems, eliminating the need for any value function. Subsequently, we implement the penalization strategy and discuss the relationship between the penalized formulation and the original problem (2).

## 2.1 PENALTY-BASED METHODS BASED ON LOWER-LEVEL DUALITY

In this subsection, we reconstruct the lower-level problem with Lagrangian function and duality. Based on this, we study the lower-level duality reformulation and propose the penalty-based method. First we introduce augmented variables $\mathbf{z}_i, i = 1, 2, ..., r$ and deduce the equivalent form of LL problem of (2),

$$\min_{\mathbf{x}, \mathbf{z}_i} l(\mathbf{x}) + \sum_{i=1}^{r} \lambda_i R_i(\mathbf{z}_i) \quad \text{s.t. } \mathbf{x} = \mathbf{z}_i. \tag{3}$$

Since $l, R_i$ are convex and the constraints are affine, strong duality holds under Slater's condition. If $\text{ri}(\text{dom } l \cap (\cap_{i=1}^{r} \text{dom } R_i)) \neq \emptyset$, then (3) is equivalent to its Lagrangian dual problem:

$$-\min_{\boldsymbol{\rho}} \max_{\mathbf{x}, \mathbf{z}_i} -l(\mathbf{x}) - \sum_{i=1}^{r} \lambda_i R_i(\mathbf{z}_i) - \sum_{i=1}^{r} \boldsymbol{\rho_i}^T (\mathbf{x} - \mathbf{z}_i),$$

where $\boldsymbol{\rho}_i$ is are Lagrangian multipliers associated with constraint $\mathbf{x} = \mathbf{z}_i$. The above problem can be further simplified with definition of conjugate functions as,

$$\max_{\boldsymbol{\rho}} -l^*(-\sum_{i=1}^{r} \boldsymbol{\rho_i}) - \sum_{i=1}^{r} \lambda_i R_i^*(-\frac{\boldsymbol{\rho}_i}{\lambda_i}). \tag{4}$$

Meanwhile, the constraint of (2) is equivalent to

$$
\begin{aligned}
l(\mathbf{x}) + \sum_{i=1}^{r} \lambda_i R_i(\mathbf{x}) &\overset{(a)}{\leq} \min_{\mathbf{x}}\{l(\mathbf{x}) + \sum_{i=1}^{r} \lambda_i R_i(\mathbf{x})\} \\
&\overset{(b)}{=} \max_{\boldsymbol{\rho}} -l^*(-\sum_{i=1}^{r} \boldsymbol{\rho_i}) - \sum_{i=1}^{r} \lambda_i R_i^*(-\frac{\boldsymbol{\rho}_i}{\lambda_i}),
\end{aligned}
\tag{5}
$$

where, $(a)$ utilizes the value function of the lower-level problem, which is widely used in relevant literature of BLO Liu et al. (2021a; 2023), $(b)$ is from the equivalence of (3)-(4). Dropping the $\max$ operator, we obtain that the lower-level problem of (2) can be replaced by the inequality constraint,

$$l(\mathbf{x}) + \sum_{i=1}^{r} \lambda_i R_i(\mathbf{x}) + l^*(-\sum_{i=1}^{r} \boldsymbol{\rho_i}) + \sum_{i=1}^{r} \lambda_i R_i^*(\frac{\boldsymbol{\rho_i}}{\lambda_i}) \leq 0,$$

and obtain the reformulation for (2):

$$
\begin{aligned}
\min_{\mathbf{x}, \boldsymbol{\lambda}, \boldsymbol{\rho}} \quad & L(\mathbf{x}) \\
\text{s.t.} \quad & l(\mathbf{x}) + \sum_{i=1}^{r} \lambda_i R_i(\mathbf{x}) + l^*(-\sum_{i=1}^{r} \boldsymbol{\rho_i}) + \sum_{i=1}^{r} \lambda_i R_i^*(\frac{\boldsymbol{\rho_i}}{\lambda_i}) \leq 0.
\end{aligned}
\tag{6}
$$

Note that it is independent of any implicit value function, but rather utilizes the conjugate of the atom functions in the lower-level problem. Naturally, the validity of (6) depends on the following assumption.

**Assumption 2.1.** $l$ and $R_i, i = 1, 2, ..., r$ in the lower-level problem of (2) possess explicit conjugate functions.

The fulfillment of Assumption 2.1 is straightforward to ensure. Indeed, the loss functions in most real-world problems have closed-form conjugate functions, including least squares, hinge loss and logarithmic functions. Similarly, the norm terms $R_i(\cdot)$ also share this property, where we denote $R_i^*(\cdot) = \|\cdot\|^*$ as the conjugate norm of $R_i$. In this case, we observe that $R_i^*(\frac{\boldsymbol{\rho}_i}{\lambda_i}) = 0$ provided the condition $\|\boldsymbol{\rho}_i\|_* \leq \lambda_i$ holds (Boyd & Vandenberghe, 2004). Meanwhile, with introducing an auxiliary variables $r_i$ satisfying $R_i(\mathbf{x}) \leq r_i$, the constraint of (6) is equivalent to

$$
\begin{aligned}
& l(\mathbf{x}) + l^*(-\sum_{i=1}^{r} \boldsymbol{\rho}_i) + \sum_{i=1}^{r} \lambda_i r_i \leq 0. \\
& R_i(\mathbf{x}) \leq r_i, \|\boldsymbol{\rho}_i\|_* \leq \lambda_i, i = 1, 2, ..., r.
\end{aligned}
\tag{7}
$$

Consequently, (6) is equivalent to the following problem,

$$
\begin{aligned}
\min_{\mathbf{x}, \boldsymbol{\lambda}, \boldsymbol{\rho}, \mathbf{r}} \quad & L(\mathbf{x}) \\
\text{s.t.} \quad & l(\mathbf{x}) + l^*(-\sum_{i=1}^{r} \boldsymbol{\rho}_i) + \sum_{i=1}^{r} \lambda_i r_i \leq 0. \\
& R_i(\mathbf{x}) \leq r_i, \|\boldsymbol{\rho}_i\|_* \leq \lambda_i, i = 1, 2, ..., r.
\end{aligned}
\tag{8}
$$

We summarize the first inequality constraint of (8) as a penalty term

$$
p(\mathbf{x}, \boldsymbol{\lambda}, \boldsymbol{\rho}, \mathbf{r}) = l(\mathbf{x}) + l^*(-\sum_{i=1}^{r} \boldsymbol{\rho}_i) + \sum_{i=1}^{r} \lambda_i r_i,
\tag{9}
$$

and employ penalization strategy to handle (8). Then we can rewrite (8) with a penalty constant $\beta$ as follows,

$$
\begin{aligned}
\min_{\mathbf{x}, \boldsymbol{\lambda}, \boldsymbol{\rho}, \mathbf{r}} \quad & L(\mathbf{x}) + \beta p(\mathbf{x}, \boldsymbol{\lambda}, \boldsymbol{\rho}, \mathbf{r}). \\
\text{s.t.} \quad & R_i(\mathbf{x}) \leq r_i, \|\boldsymbol{\rho}_i\|_* \leq \lambda_i, i = 1, 2, ..., r.
\end{aligned}
\tag{10}
$$

Thus, we have fully converted the hyperparameter optimization (2) into a single-level formulation (10). Although the introduced variable $\boldsymbol{\rho}_i$ has the same dimension as $\mathbf{x}$, it does not affect the whole scale and complexity.

## 2.2 EQUIVALENCE BETWEEN PENALIZED AND PRIMAL PROBLEM

In this subsection, we discuss the relationship between (2) and (10) from the perspective of duality. We first introduce corresponding assumptions for 2 as follows.

**Assumption 2.2.** $L(\mathbf{x})$ is $L_0$-Lipschitz continuous.

**Assumption 2.3.** $l(\mathbf{x})$ is $(1/\alpha_l)$-strongly convex and $l_1$-smooth.

**Assumption 2.4.** For any given $\mathbf{x}$, the optimal solution set of lower-level problem in (2) denoted as $L_{\text{opt}}(\boldsymbol{\lambda})$ is closed and non-empty.

Besides Assumption 2.2, we note that the norm terms $R_i(\mathbf{x})$ are convex but potentially nonsmooth, which implies that the lower-level problem is convex and nonsmooth in $\mathbf{x}$. Regarding Assumptions 2.2 and 2.3, the conjugate function $l^*$ is $\alpha_l$-smooth (Theorem 5.26 in Beck (2017)). Subsequently, the penalty term $p(\mathbf{x}, \boldsymbol{\lambda}, \boldsymbol{\rho}, \mathbf{r})$ is differentiable and $(l_1 + \alpha_l + 1)$-smooth. The above assumptions are prevalent and commonly satisfied in practical applications. From (3)-(8), we know that (2) can be reformulated into (8). From the KKT conditions of (3), we first analyze $\boldsymbol{\rho}_i, i = 1, 2, ..., r$ in (6) and obtain the following lemma.

**Lemma 2.5.** *If $\mathbf{x}_{\boldsymbol{\lambda}}$ is an optimal solution of the lower-level problem of (2), then there exists the unique multiplier $\boldsymbol{\rho}_i^*$ and $\mathbf{z}_i^* = \mathbf{x}_{\boldsymbol{\lambda}}$ such that $(\mathbf{x}_{\boldsymbol{\lambda}}, \mathbf{z}_i^*, \boldsymbol{\rho}_i^*)$ is a KKT point of (3).*

According to KKT condition of we recover that $\boldsymbol{\rho}_i^*$ in Lemma 2.5 satisfies that

$$
\sum_{i=1}^{r} \boldsymbol{\rho}_i^* = -\nabla l(\mathbf{x}_{\boldsymbol{\lambda}}), \ \boldsymbol{\rho}_i^* \in \lambda_i \partial R_i(\mathbf{x}_{\boldsymbol{\lambda}}), i = 1, 2, ..., r,
\tag{11}
$$

which implies that the KKT point of (3) is also the stationary point of the lower-level problem of (2). Note that the penalty term $p(\mathbf{x}, \boldsymbol{\lambda}, \boldsymbol{\rho}, \mathbf{r})$ is derived from duality of lower-level problem, so we summarize the property of $p(\mathbf{x}, \boldsymbol{\lambda}, \boldsymbol{\rho}, \mathbf{r})$ regulating $\|\mathbf{x} - \mathbf{x}_{\boldsymbol{\lambda}}\|^2$ as follows.

**Lemma 2.6.** *Suppose Assumption 2.3 and 2.4 hold, then it holds that $p(\mathbf{x}, \boldsymbol{\lambda}, \boldsymbol{\rho}, \mathbf{r}) \geq \frac{\alpha_l}{2} \|\mathbf{x} - \mathbf{x}_{\boldsymbol{\lambda}}\|^2 \geq 0$ for any given $\mathbf{x}, \boldsymbol{\lambda}, \boldsymbol{\rho}, \mathbf{r}$. In addition, $p(\mathbf{x}, \boldsymbol{\lambda}, \boldsymbol{\rho}, \mathbf{r}) = 0$ if and only if $\mathbf{x} \in L_{opt}(\boldsymbol{\lambda})$.*

Based on Lemma 2.5, we further derive the equivalence between bilevel form (2) and the constrained problem (6) as follows.

**Proposition 2.7.** *If $(\mathbf{x}^*, \boldsymbol{\lambda}^*)$ is a global optimal solution for (2), and $\boldsymbol{\rho}_i^*$ is defined as in (11), then $(\mathbf{x}^*, \boldsymbol{\lambda}^*, \boldsymbol{\rho}_i^*)$ is global optimal solution for (6).*

From Proposition 2.7, we can further recognize the equivalence between the primal problem (2) and (8). As a result, we now redirect our focus to investigating relationship between (8) and (10).

Due to the non-negativity of the penalty term $p(\mathbf{x}, \boldsymbol{\lambda}, \boldsymbol{\rho}, \mathbf{r})$, we find that there is no interior points in the feasible region of (6)(8), in the sense that the constraint contradicts any standard regularity condition. Therefore, we consider the following $\epsilon$-approximate problem for (6)(8) and discuss the equivalence between it and the penalty problem (10),

$$
\begin{aligned}
\min_{\mathbf{x}, \boldsymbol{\lambda}, \boldsymbol{\rho}, \mathbf{r}} \quad & L(\mathbf{x}) \\
\text{s.t.} \quad & p(\mathbf{x}, \boldsymbol{\lambda}, \boldsymbol{\rho}, \mathbf{r}) \leq \epsilon. \\
& R_i(\mathbf{x}) \leq r_i, \|\boldsymbol{\rho_i}\|_* \leq \lambda_i, i = 1, 2, ..., r.
\end{aligned} \tag{12}
$$

Leveraging Lemma 2.6, we establish the relationship between global optimal solutions of (10) and 12 in Proposition 2.8, which is inspired by Shen & Chen (2023).

**Proposition 2.8.** *Suppose Assumption 2.3 and 2.4 hold. For any $\epsilon_p > 0$, the global optimal solution of (2) is also an $\epsilon_p$-approximation optimal solution of the penalized problem (10) with $\beta > \beta^* = \frac{l_0^2 \alpha_l}{8\epsilon_p}$. Conversely, the $\epsilon_1$-global solution of (10) with $\beta > \beta^*$ is a global optimal solution for $\epsilon$-approximate problem (12) with $0 \leq \epsilon \leq (\epsilon_p + \epsilon_1)/(\beta - \beta^*)$.*

In summary, we confirm the relationship between the penalized problem (10) and primal problem (2). Subsequently, we illustrate the proximity between the optimal value of (10) and (2).

**Theorem 2.9.** *Suppose that Assumptions 2.2, 2.3 and 2.4 hold. If $(\mathbf{x}_\epsilon^*, \boldsymbol{\lambda}_\epsilon^*, \boldsymbol{\rho}_\epsilon^*, \mathbf{r}_\epsilon^*)$ is $\epsilon$-optimal solution of the penalized problem (10), then we obtain that $|L(\mathbf{x}_\epsilon^*) - L(\mathbf{x}^*)| \leq \mathcal{O}(\epsilon)$, where $\mathbf{x}^*$ with an optimal $\boldsymbol{\lambda}^*$ attains the minimum of (2).*

We provide the related proofs in Appendix A. The primary challenges in solving (10) arise from its nonsmooth and nonconvex properties. To address these, we explore first-order algorithms to solve the penalized problem (10), cleverly leveraging the structure of (2) and (10).

## 3 SOLVING THE PENALTY FORMULATIONS

In this section, we propose our main algorithm grounded in penalty-based problem (10). For convenience, we denote $\mathbf{z} = (\mathbf{x}, \boldsymbol{\lambda}, \boldsymbol{\rho}, \mathbf{r})$. We then introduce the constraint sets for each $i$ as follows,

$$
\mathcal{R}_i \triangleq \{\mathbf{z} | R_i(\mathbf{x}) \leq r_i\}, \quad \mathcal{R}_i^* \triangleq \{\mathbf{z} | \|\boldsymbol{\rho_i}\|_* \leq \lambda_i\}. \tag{13}
$$

A natural approach to manage the constraints of (10) is through projection onto $\mathcal{R}_i$ and $\mathcal{R}_i^*$. To proceed, we introduce the following assumption regarding $\mathcal{R}_i$ and $\mathcal{R}_i^*$.

**Assumption 3.1.** For the constraint sets $\mathcal{R}_i, i = 1, 2, ..., r$, each individual set among these $r$ sets can be easy to project, implying that the corresponding indicator functions $\mathcal{I}_{\mathcal{R}_i}(\mathbf{z})$ are proximal-friendly for each $i$, respectively.

From Moreau decomposition theorem (Theorem 6.44 in Beck (2017)), we know that each individual set $\mathcal{R}_i^*$ and corresponding indicator functions $\mathcal{I}_{\mathcal{R}_i^*}(\mathbf{z})$ satisfy the same property described in Assumption 3.1 for $\mathcal{R}_i$. Assumption 3.1 holds for common norm terms. Even if the constraints of (10) are in conic form, the corresponding projections still have close-form solutions for each $i$. We explain the specific analytic solutions of projection in Appendix C.

However, significant differences exist between the two groups of constraints related to norms and their conjugate, as the constraints $R_i(\mathbf{x}) \leq r_i$ are all related to the same variable $\mathbf{x}$ while the constraints $\|\boldsymbol{\rho_i}\|_* \leq \lambda_i$ pertain to entirely different variables $\boldsymbol{\rho}_i$. Consequently, the projection process for $\cap_{i=1}^r \mathcal{R}_i$ will involve complicated interactions among the feasible domain of each constraint $R_i(\mathbf{x}) \leq r_i$. In other words, the constraint sets $\mathcal{R}_i^*$ are mutually separated, which means that $\cap_{i=1}^r \mathcal{R}_i^*$ is easy to project. Accordingly, the projection onto $\cap_{i=1}^r \mathcal{R}_i$ is hard to directly computed and its indicator function is generally proximal-unfriendly.

Although relevant full projection algorithms for composite constraints are explored by Li et al. (2020); Liu & Liu (2017), these algorithms necessitate additional iterative loop and produce inexact results. Thus, the integration of these full projections with first-order algorithms can lead to divergence and a notable decrease in efficiency. Therefore, we need to consider splitting the mixed constraint sets $\cap_{i=1}^r \mathcal{R}_i$. In the specific scenario of problem (2) with a single regularizer, the obstacles are rendered unnecessary.

Therefore, we introduce the first-order algorithm for a single regularizer ($r = 1$) as a special case in subsection 3.1, while the algorithm for problems requiring multiple norm regularization terms ($r > 1$) is presented in subsection 3.2.

### 3.1 SINGLE REGULARIZATION TERM

In this subsection, we explore the algorithm for (2) with a single regularization term $R_1(\mathbf{x})$. Consequently, (10) simplifies to the following formulation:

$$\min_{\mathbf{x},\boldsymbol{\lambda},\boldsymbol{\rho},\mathbf{r}} \quad L(\mathbf{x}) + \beta p(\mathbf{x}, \boldsymbol{\lambda}, \boldsymbol{\rho}, \mathbf{r}).$$
$$\text{s.t.} \quad R_1(\mathbf{x}) \leq r_1, \|\boldsymbol{\rho}\|_* \leq \lambda_1, \tag{14}$$

where $p(\mathbf{x}, \boldsymbol{\lambda}, \boldsymbol{\rho}, \mathbf{r}) = l(\mathbf{x}) + l^*(-\boldsymbol{\rho}) + \lambda_1 r_1$. We adopt the notations $\mathbf{z} = (\mathbf{x}, \boldsymbol{\lambda}, \boldsymbol{\rho}, \mathbf{r})$ and define $\mathcal{R}_1, \mathcal{R}_1^*$ as in (13).

**Definition 3.2.** A function $f$ is called $w$-weakly convex for some $w \geq 0$ if $f(\cdot) + \frac{w}{2}\|\cdot\|^2$ is convex.

It is noteworthy that the bilinear term $\lambda_1 r_1$ is 1-weakly convex and 1-smooth with respect to $\mathbf{z}$.

**Lemma 3.3.** $L(\mathbf{x}) + \beta p(\mathbf{z})$ is $l_p$-smooth in $\mathbf{z}$ with $l_p \triangleq l_1 + \beta(l_1 + \alpha_l + 1)$.

The above results can be directly computed under Assumptions 2.2 and 2.3. Meanwhile, the sets $\mathcal{R}_1$ satisfies Assumption 3.1 and it is separated from $\mathcal{R}_1^*$. Therefore, $\mathcal{R}_1 \cap \mathcal{R}_1^*$ is projected-friendly and (14) can be minimized with projected gradient descent. We summarize our first-order algorithm for (14) in Algorithm 1. In line 1, $\mathbf{x}^0$ is initialized by solving lower-level problem $\min_{\mathbf{x}}\{l(\mathbf{x}) + \lambda_1 R_1(\mathbf{x})\}$ with given $\lambda_1^0$ and we set $\mathbf{r}^0 = R_1(\mathbf{x}^0), \boldsymbol{\rho}^0 = -\nabla l(\mathbf{x}^0)$. In this setting, we ensure the feasibility of problem (14). In line 3, the iterative first-order method is performed for problem (14) accompanied by the projection onto $\mathcal{R}_1 \cap \mathcal{R}_1^*$. With the fixed penalty parameter $\beta$, we set the step size $\eta \leq 2/l_p$ and $l_p$ is computed in Lemma 3.3, which ensures consistent progression throughout the iterations. In line 4, we choose the stopping criterion with the results of two iterative points are sufficiently close, i.e., $\|\mathbf{z}^{k+1} - \mathbf{z}^k\| \leq \text{tol}$.

---

**Algorithm 1** First-order Methods for Penalized Problem (14)

1: Initialize $\boldsymbol{\lambda}^0$ and $\mathbf{x}^0, \boldsymbol{\rho}^0, \mathbf{r}^0$, constants $\beta, \eta$.
2: **for** $k = 0, 1, 2, ..., K$ **do**
3:     Update $\mathbf{z}^{k+1} = \text{proj}_{\mathcal{R}_1 \cap \mathcal{R}_1^*}\{\mathbf{z}^k - \eta[\nabla_{\mathbf{z}}(L(\mathbf{x}^k) + \beta p(\mathbf{z}^k))]\}$.
4:     **if** Termination criteria is met. **then**
5:         Stop.
6:     **end if**
7: **end for**

---

*Remark* 3.4. We define an indicator function as $g_1(\mathbf{z}) = \mathcal{I}_{\mathcal{R}_1 \cap \mathcal{R}_1^*}(\mathbf{z})$. The iteration 3 in Algorithm 1 can be described as the process of finding an approximate optimal solution of (14).

Since the reformulation (6) involves no implicit value functions related to the lower-level problem of (2), Algorithm 1 does not require an iterative loop for finding the optimal solution $\mathbf{x}_{\boldsymbol{\lambda}}$ of lower-level problem of (2) or the dual multiplier $\boldsymbol{\rho}^*$. Therefore, Algorithm 1 is equipped with a single loop for $\mathbf{z}$, which fully centers on the variables $(\mathbf{x}, \boldsymbol{\lambda}, \boldsymbol{\rho}, \mathbf{r})$ in problem (14).

In this case, we obtain the sufficient decrease and convergence results of Algorithm 1 as follows.

**Lemma 3.5.** *Assume $L(\mathbf{x})$ and $p(\mathbf{z})$ are bounded below. For $k \in \mathbb{N}$ and $\{\mathbf{z}^k\}$ generated from Algorithm 1 with penalty parameter $\bar{\beta}$, we have $L(\mathbf{x}^{k+1}) + \bar{\beta}p(\mathbf{z}^{k+1}) \leq L(\mathbf{x}^k) + \bar{\beta}p(\mathbf{z}^k)$. In addition, the sequence $\{\mathbf{z}^k\}$ satisfies that $\lim_{k\to\infty} \|\mathbf{z}^{k+1} - \mathbf{z}^k\| = 0$.*

**Theorem 3.6.** *Assume $L(\mathbf{x})$ and $p(\mathbf{z})$ are bounded below. Based on Lemma 3.5, any limit point of $\{\mathbf{z}^k\}$ is a stationary point of (14).*

The proofs of Lemma 3.5 and Theorem 3.6 are provided in Appendix B. The convergence results in this case follow from Beck & Teboulle (2009; 2010), which introduce the analysis of proximal gradient method. In summary, Algorithm 1 addresses the primal problem (2) with single regularization term by applying the penalized problem in the form of (14). It also inspires the resolution of the cases involving multiple regularization terms.

### 3.2 DOUBLE REGULARIZATION TERMS

In this subsection, we focus on the algorithm design for (2) involving multiple regularization terms. For convenience, we present the case with double regularization terms in the main text, while the algorithm for addressing (2) with more regularization terms and correspondingly results are provided in Appendix B.5. For this scenario, (10) simplifies to the following formulation:

$$\min_{\mathbf{x},\boldsymbol{\lambda},\boldsymbol{\rho},\mathbf{r}} \quad L(\mathbf{x}) + \beta p(\mathbf{x},\boldsymbol{\lambda},\boldsymbol{\rho},\mathbf{r}).$$
$$\text{s.t.} \quad R_i(\mathbf{x}) \leq r_i, \|\boldsymbol{\rho}_i\|_* \leq \lambda_i, i = 1, 2, \tag{15}$$

where $p(\mathbf{x},\boldsymbol{\lambda},\boldsymbol{\rho},\mathbf{r}) = l(\mathbf{x}) + l^*(-\boldsymbol{\rho}_1 - \boldsymbol{\rho}_2) + \lambda_1 r_1 + \lambda_2 r_2$. We adopt the notations $\mathbf{z} = (\mathbf{x},\boldsymbol{\lambda},\boldsymbol{\rho},\mathbf{r})$ and $\mathcal{R}_i, \mathcal{R}_i^*, i = 1, 2$ defined in (13). From Assumption 3.1, we know that $\mathcal{R}^* \triangleq \mathcal{R}_1^* \cap \mathcal{R}_2^*$ is projected-friendly, so we merely need to perform variable decomposition for $\mathcal{R}_1 \cap \mathcal{R}_2$. We define $g_i(\mathbf{z}) \triangleq \mathcal{I}_{\mathcal{R}_i \cap \mathcal{R}^*}(\mathbf{z}), i = 1, 2$. Under this conditions, (15) can be rewritten as the following equivalent form,

$$\min_{\mathbf{z}} \quad L(\mathbf{x}) + \beta p(\mathbf{z}) + g_1(\mathbf{z}) + g_2(\mathbf{z}). \tag{16}$$

Motivated by (3), we introduce an auxiliary variable $\mathbf{u}$ as follows,

$$\min_{\mathbf{z}} \quad L(\mathbf{x}) + \beta p(\mathbf{z}) + g_1(\mathbf{z}) + g_2(\mathbf{u})$$
$$\text{s.t.} \quad \mathbf{z} = \mathbf{u}. \tag{17}$$

The augmented Lagrangian function of problem (17) is

$$\begin{aligned}
\mathcal{L}_\gamma(\mathbf{z},\mathbf{u},\boldsymbol{\mu}) &= L(\mathbf{x}) + \beta p(\mathbf{z}) + g_1(\mathbf{z}) + g_2(\mathbf{u}) + \langle \boldsymbol{\mu}, \mathbf{u} - \mathbf{z} \rangle + \frac{\gamma}{2}\|\mathbf{u} - \mathbf{z}\|^2 \\
&= L(\mathbf{x}) + \beta p(\mathbf{z}) + g_1(\mathbf{z}) + g_2(\mathbf{u}) + \frac{\gamma}{2}\|\mathbf{u} - \mathbf{z} + \frac{\boldsymbol{\mu}}{\gamma}\|^2 - \frac{\|\boldsymbol{\mu}\|^2}{2\gamma}.
\end{aligned}$$

Now, we naturally employ Alternating Direction Method of Multipliers (ADMM) to solve (17), which cyclically update $\mathbf{u}, \mathbf{z}, \boldsymbol{\mu}$ by solving the $\mathbf{u}$- and $\mathbf{z}$-subproblems and adopt a dual ascent step for $\boldsymbol{\mu}$. We summarize the iterations in Algorithm 2. In line 1, $\mathbf{x}^0$ is initialized by solving lower-level problem $\min_{\mathbf{x}}\{l(\mathbf{x}) + \lambda_1 R_1(\mathbf{x}) + \lambda_2 R_2(\mathbf{x})\}$ with given $\boldsymbol{\lambda}^0$ and we set $r_i^0 = R_i(\mathbf{x}^0)$. In line 3, we add a proximal term due to the weakly-convex term $\lambda_i r_i, i = 1, 2$ with a constant $t$. In line 4, $\mathbf{u}$-subproblem takes the form of direct projection onto $\mathcal{R}_2$. Under Assumption 3.1, we assume that $\mathbf{u}$-subproblem can be solved exactly in each iteration.

---

**Algorithm 2** ADMM Framework for Problem (15)

1: Initialize $\boldsymbol{\lambda}^0$ and $\mathbf{x}^0, \boldsymbol{\rho}^0, \mathbf{r}^0, \mathbf{u}^0 = (\mathbf{x}^0, \boldsymbol{\lambda}^0, \boldsymbol{\rho}^0, \mathbf{r}^0)$, constants $\beta, \gamma$ and $t$.
2: **for** $k = 0, 1, 2, ...$ **do**
3:    $\mathbf{z}^{k+1} = \arg\min_{\mathbf{z}}\left\{L(\mathbf{x}) + \beta p(\mathbf{z}) + g_1(\mathbf{z}) + \frac{\gamma}{2}\|\mathbf{u}^k - \mathbf{z} + \frac{\boldsymbol{\mu}^k}{\gamma}\|^2 + \frac{t}{2}\|\mathbf{z} - \mathbf{z}^k\|^2\right\}.$
4:    $\mathbf{u}^{k+1} = \arg\min_{\mathbf{u}}\left\{g_2(\mathbf{u}) + \frac{\gamma}{2}\|\mathbf{u} - \mathbf{z}^{k+1} + \frac{\boldsymbol{\mu}^k}{\gamma}\|^2\right\}.$
5:    $\boldsymbol{\mu}^{k+1} = \boldsymbol{\mu}^k + \gamma(\mathbf{u}^{k+1} - \mathbf{z}^{k+1}).$
6: **end for**

---

According to Definition 3.2, we control the proximal coefficient with $t > \alpha_d - \gamma$ where $\alpha_d \triangleq \frac{\beta}{2} - (1 + \beta)\alpha_l - \gamma$, then we describe the property of $\mathbf{z}$-subpoblem in the following lemma.

**Lemma 3.7.** *Suppose Assumptions 2.2 and 2.3 hold. The $\mathbf{z}$-subproblem in line 3 of Algorithm 2 enjoys $(t - \alpha_d)$-strongly convex property, while the objective function is $l_d$-smooth with $l_d \triangleq \gamma + t + l_1 + \beta(l_1 + \alpha_l + 1)$.*

The above results is obtained from direct computation under Assumptions 2.2 and 2.3. For $\mathbf{z}$-subproblem in line 3, $g_1(\mathbf{z})$ is indicator function and the problem can be expressed in the following form

$$\mathbf{z}^{k+1} = \arg\min_{\mathbf{z} \in \mathcal{R}_1 \cap \mathcal{R}^*}\left\{L(\mathbf{x}) + \beta p(\mathbf{z}) + \frac{\gamma}{2}\|\mathbf{u}^k - \mathbf{z} + \frac{\boldsymbol{\mu}^k}{\gamma}\|^2 + \frac{t}{2}\|\mathbf{z} - \mathbf{z}^k\|^2\right\}, \tag{18}$$

which can be solved with projected gradient descent in the form of Algorithm 1 with a constant step size $\eta \leq \frac{1}{l_d}$. The projected gradient descent for the $z$-subproblem includes an additional proximal term compared to Algorithm 1. Note that (18) is strongly convex and smooth from Lemma 3.7, then we can derive the complexity results for finding an $\epsilon_k$-optimal solution for $z$-subproblem in $k$-th iteration of Algorithm 2.

**Lemma 3.8.** *In $k$-th iteration of Algorithm 2, an $\epsilon_k$-optimal solution $z^{k+1}$ is generated in $\mathcal{O}(\frac{l_d}{t-\alpha_d} \log(\frac{1}{\epsilon_k}))$ projected gradient descent oracles.*

The results of complexity of inner iterations utilize the conclusive findings in Bubeck et al. (2015). Then we make the assumptions concerning $z$-subproblem and $\mu$.

**Assumption 3.9.** The sequence $\{\epsilon_k\}$ satisfies $\sum\limits_{k=1}^{\infty} \epsilon_k < \infty$.

**Assumption 3.10.** The sequence $\{\mu^k\}$ is bounded and satisfies $\sum\limits_{k=1}^{\infty} \|\mu^k - \mu^{k+1}\|^2 < \infty$.

Assumption 3.9 is introduced by Wang et al. (2019) and Assumption 3.10 is popularly employed in ADMM approaches Xu et al. (2012); Bai et al. (2021); Shen et al. (2014); Cui et al. (2024). Based on Assumptions 3.9 and 3.10, we propose the convergence result for Algorithm 2 in Theorem 3.11.

**Theorem 3.11.** *Algorithm 2 can find an $\epsilon$-KKT point $(z^{k+1}, u^{k+1}, \mu^{k+1})$ of (17) within $\mathcal{O}(1/\epsilon^2)$ iterations.*

From Theorem 3.11, we further conclude that Algorithm 2 finds an $\epsilon$-KKT point of (17) within $\mathcal{O}(1/\epsilon^2)$ iterations. we provide the detailed proofs and extension to problem (2) with multiple regularizers in Appendix B.

## 4 NUMERICAL EXPERIMENTS

In this section, we conduct experiments to compare LDPM with existing algorithms for hyperparameter optimization on synthetic data and real datasets, respectively. In specific, we mainly compare our LDPM with grid search, random search, TPE (Bergstra et al., 2013), IJGO (Feng & Simon, 2018), VF-iDCA (Gao et al., 2022), LDMMA (Chen et al., 2024), GAFFA (Yao et al., 2024a). All experiments are performed on a computer with Intel(R) Core(TM) i7-10710U CPU @ 1.10GHz 1.61 GHz and 16.00 GB memory. The code is implemented using Python 3.9. We consider hyperparameter optimization for elastic net and (sparse) group lasso. In this section, we present part of the experimental results on synthetic data, with additional results and detailed descriptions of the data generation and parameters for several methods included in Appendix D.

### 4.1 SPARSE GROUP LASSO

We conduct experiments with different data scales and report results in Figure 1. The results of the search methods and Bayesian method (TPE) are not presented in Figure 1 due to its lower efficiency and instability. We have included the specific numerical results in tabular form in Appendix D.1. We observe that LDPM consistently outperforms other algorithms in terms of computational efficiency. As the data scale increases, the superiority of our approach becomes increasingly evident, demonstrating the advantages of LDPM in large-scale hyperparameter optimization. In contrast, gradient-free methods exhibit significant instability when handling numerous hyperparameters, while IGJO converges slowly and demands substantial computational resources. Our iteration process is independent of any solvers, allowing it to outperform LDMMA and VF-iDCA, both of which rely on specific solvers for their iterative subproblems.

### 4.2 ELASTIC NET

The numerical results on elastic net are reported in Figure 2. Overall, LDPM achieves the highest solution quality in the shortest running time on this problem model. Similar to Section 4.1, the results of the search method and Bayesian method are not presented in the figure. Instead, we have included other results in tabular form in the Appendix D.1. Overall, LDPM achieves the lowest test error with

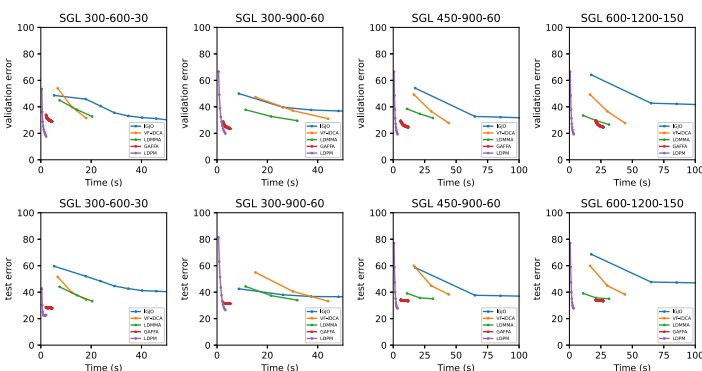

Figure 1: Comparison of the algorithms on Group Lasso problem for synthetic datasets in different scales

significantly lower time costs, particularly in large-scale data scenarios. While the gradient-based method IGJO demonstrates slightly better accuracy and efficiency and its convergence is notably slow as illustrated in the figure. Meanwhile, VF-iDCA and LDMMA maintain consistently low validation errors across all experiments. However, both algorithms suffer from overfitting, resulting in increased test errors as the iterations progress.

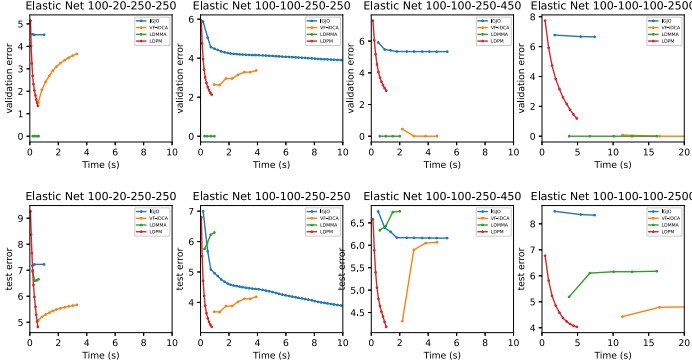

Figure 2: Comparison of the algorithms on Elastic Net problem for synthetic datasets in different scales

We present other experimental results in the form of figures and tables in Appendix D.1 and D.2, demonstrating the robustness and applicability of our algorithm. Notably, our algorithm does not utilize any open-source libraries like CVXPY or commercial optimization solvers, such as MOSEK, which are typically employed in many hyperparameter optimization algorithms.

## 5 CONLUSIONS

This paper addresses hyperparameter optimization in the context of nonsmooth regularizers by proposing a novel penalty method based on lower-level duality (LDPM). Our approach applies penalization to a single-level reformulation, eschewing any implicit value function and instead utilizing the conjugates of atomic functions. We effectively solve the subproblems within this penalization framework using fully first-order methods, including proximal techniques and the alternating direction method of multipliers, while maintaining simplicity by avoiding complex off-the-shelf solvers or high-complexity iterations. Theoretical analyses substantiate the convergence of our method. Our numerical experiments, conducted on both synthetic and real-world datasets, demon-

strate that LDPM consistently outperforms existing methodologies, with its advantages particularly pronounced in large-scale scenarios. Looking ahead, we aim to explore nonsmooth loss functions and develop more general algorithms from a stochastic perspective, thereby broadening the applicability and impact of our approach.

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
