# A    PROOF IN SECTION 2

In this subsection, we provide the proof for the results concerning the penalty framework in Section 2. First, Lemma 2.5 and Proposition 2.7 hold under the strong duality of (3) (Boyd & Vandenberghe, 2004). We present detailed proofs for Lemma 2.6, Proposition 2.8 and Theorem 2.9 in the subsequent discussion.

## A.1    PROOF OF LEMMA 2.6

*Proof.* We restate the lower-level problem of (2) as follows,

$$\min_{\mathbf{x}}\{l(\mathbf{x}) + \sum_{i=1}^{r}\lambda_i R_i(\mathbf{x})\}. \tag{19}$$

We first analyze the maximum and minimum in (5). From Lemma 2.5 and Proposition 2.7, we know that the $\max$ operator with respect to $\boldsymbol{\rho}$ is achieved at $\boldsymbol{\rho}_i^*$ defined in (11). Meanwhile, the $\min$ operator of $\mathbf{x}$ occurs at $\mathbf{x} = \mathbf{x}_{\boldsymbol{\lambda}}$. According to the definition of $p(\mathbf{x}, \boldsymbol{\rho}, \boldsymbol{\lambda}, \mathbf{r})$, we deduce that

$$
\begin{aligned}
p(\mathbf{x}, \boldsymbol{\rho}, \boldsymbol{\lambda}, \mathbf{r}) \quad &= \quad l(\mathbf{x}) + l^*(-\sum_{i=1}^{r}\boldsymbol{\rho}_i) + \sum_{i=1}^{r}\lambda_i r_i \\
&\overset{(a)}{\geq} \quad l(\mathbf{x}) + \sum_{i=1}^{r}\lambda_i R_i(\mathbf{x}) + l^*(-\sum_{i=1}^{r}\boldsymbol{\rho}_i) \\
&\overset{(b)}{\geq} \quad l(\mathbf{x}) + \sum_{i=1}^{r}\lambda_i R_i(\mathbf{x}) - \min_{\boldsymbol{\rho}}\{l^*(-\sum_{i=1}^{r}\boldsymbol{\rho}_i) + \sum_{i=1}^{r}\lambda_i R_i^*(\frac{\boldsymbol{\rho_i}}{\lambda_i})\} \\
&\overset{(c)}{=} \quad l(\mathbf{x}) + \sum_{i=1}^{r}\lambda_i R_i(\mathbf{x}) - l(\mathbf{x}_{\boldsymbol{\lambda}}) + \sum_{i=1}^{r}\lambda_i R_i(\mathbf{x}_{\boldsymbol{\lambda}}) \\
&\overset{(d)}{\geq} \quad \frac{\alpha_l}{2}\|\mathbf{x} - \mathbf{x}_{\boldsymbol{\lambda}}\|^2.
\end{aligned}
$$

In the above inequalities, $(a)$ results from the constraint $R_i(\mathbf{x}) \leq r_i$, $(b)$ is from the $\min$ operator where the $\min$ and $\max$ operators have been exchanged by adding the negative sign, $(c)$ follows from the results in (5) and $(d)$ leverages the strong convexity of $l(\mathbf{x})$ and the quadratic-growth condition established in Theorem 2 of Karimi et al. (2016). Moreover, when $\mathbf{x} = \mathbf{x}_{\boldsymbol{\lambda}}$ attains the minimum of the lower-level problem of (2), $(a)$ and $(c)$ hold as "=". Then we complete the proof. □

## A.2    PROOF OF PROPOSITION 2.8

*Proof.* For any $(\mathbf{x}, \boldsymbol{\lambda}, \boldsymbol{\rho}, \mathbf{r})$ feasible to (8), we have $L(\mathbf{x}^*) \leq L(\mathbf{x})$. From Lemma 2.6, if holds that $p(\mathbf{x}^*, \boldsymbol{\lambda}^*, \boldsymbol{\rho}^*, \mathbf{r}^*) = 0$. Let $\bar{\mathbf{x}}$ be the projection into $L_{\text{opt}}(\boldsymbol{\lambda})$ of $\mathbf{x}$, i.e., $\|\mathbf{x} - \bar{\mathbf{x}}\| = \text{dist}(\mathbf{y}, L_{\text{opt}}(\boldsymbol{\lambda}))$. Then we have

$$
\begin{aligned}
&L(\mathbf{x}) + \beta^* p(\mathbf{x}, \boldsymbol{\lambda}, \boldsymbol{\rho}, \mathbf{r}) - L(\bar{\mathbf{x}}) \\
&\geq \quad L(\mathbf{x}) - L(\bar{\mathbf{x}}) + \frac{\alpha_l \beta^*}{2}\|\mathbf{x} - \bar{\mathbf{x}}\|^2 \\
&\overset{(a)}{\geq} \quad L_0\|\mathbf{x} - \bar{\mathbf{x}}\| + \frac{\alpha_l \beta^*}{2}\|\mathbf{x} - \bar{\mathbf{x}}\|^2 \\
&\geq \quad \min_{\mathbf{t}} L_0 \mathbf{t} + \frac{\alpha_l \beta^*}{2}\mathbf{t}^2 \\
&\overset{(b)}{\geq} \quad -\epsilon_p.
\end{aligned} \tag{20}
$$

Here, $(a)$ is from the Lipschitz continuity assumption of $L(\mathbf{x})$, $(b)$ is from the fact that $L_0 \mathbf{t} + \frac{\alpha_l \beta^*}{2}\mathbf{t}^2$ attains its minimum at $\mathbf{t} = \frac{L_0}{\alpha_l \beta^*}$. Since $\bar{\mathbf{x}} \in L_{\text{opt}}(\boldsymbol{\lambda})$ is feasible to (2), we know that

$$L(\mathbf{x}) + \beta p(\mathbf{x}, \boldsymbol{\lambda}, \boldsymbol{\rho}, \mathbf{r}) - L(\bar{\mathbf{x}}) \geq L(\mathbf{x}) + \beta^* p(\mathbf{x}^*, \boldsymbol{\lambda}^*, \boldsymbol{\rho}^*, \mathbf{r}^*) - L(\mathbf{x}^*) \geq -\epsilon_p, \forall \beta \geq \beta^*.$$

Along with the fact that $p(\mathbf{x}^*, \boldsymbol{\lambda}^*, \boldsymbol{\rho}^*, \mathbf{r}^*) = 0$, we know that

$$L(\mathbf{x}^*) + p(\mathbf{x}^*, \boldsymbol{\lambda}^*, \boldsymbol{\rho}^*, \mathbf{r}^*) < L(\mathbf{x}) + \beta p(\mathbf{x}, \boldsymbol{\lambda}, \boldsymbol{\rho}, \mathbf{r}) + \epsilon_p, \forall \beta \geq \beta^*. \tag{21}$$

Therefore, we conclude that $(\mathbf{x}^*, \boldsymbol{\lambda}^*, \boldsymbol{\rho}^*, \mathbf{r}^*)$ is a $\epsilon_p$-global optimal solution of (10) with $\beta \geq \beta^*$.

On the converse, for any $(\mathbf{x}, \boldsymbol{\lambda}, \boldsymbol{\rho}, \mathbf{r})$ feasible for (10), we have $L(\mathbf{x}^*_\beta) + \beta p(\mathbf{x}^*_\beta, \boldsymbol{\lambda}^*_\beta, \boldsymbol{\rho}^*_\beta, \mathbf{r}^*_\beta) \leq L(\mathbf{x}) + \beta(\mathbf{x}, \boldsymbol{\lambda}, \boldsymbol{\rho}, \mathbf{r}) + \epsilon_1$. Substituting $(\mathbf{x}, \boldsymbol{\lambda}, \boldsymbol{\rho}, \mathbf{r}) = (\mathbf{x}^*, \boldsymbol{\lambda}^*, \boldsymbol{\rho}^*, \mathbf{r}^*)$, we deduce that

$$
\begin{aligned}
L(\mathbf{x}^*_\beta) + \beta p(\mathbf{x}^*_\beta, \boldsymbol{\lambda}^*_\beta, \boldsymbol{\rho}^*_\beta, \mathbf{r}^*_\beta) & \leq & L(\mathbf{x}^*) + \epsilon_1 \\
& \overset{(c)}{\leq} & L(\mathbf{x}^*_\beta) + \beta p(\mathbf{x}^*_\beta, \boldsymbol{\lambda}^*_\beta, \boldsymbol{\rho}^*_\beta, \mathbf{r}^*_\beta) + \epsilon + \epsilon_1.
\end{aligned}
\tag{22}
$$

where $(c)$ follows from the inequality relation in (20). Therefore, we have $p(\mathbf{x}^*_\beta, \boldsymbol{\lambda}^*_\beta, \boldsymbol{\rho}^*_\beta, \mathbf{r}^*_\beta) \leq (\epsilon + \epsilon_1)/(\beta - \beta^*)$. Define $\epsilon_\beta = p(\mathbf{x}^*_\beta, \boldsymbol{\lambda}^*_\beta, \boldsymbol{\rho}^*_\beta, \mathbf{r}^*_\beta)$, then we have $\epsilon_\beta \leq (\epsilon + \epsilon_1)/(\beta - \beta^*)$. Then for any $(\mathbf{x}, \boldsymbol{\lambda}, \boldsymbol{\rho}, \mathbf{r})$ feasible for (12) with $\epsilon = \epsilon_\beta$, it holds that $L(\mathbf{x}^*_\beta) + \beta p(\mathbf{x}^*_\beta, \boldsymbol{\lambda}^*_\beta, \boldsymbol{\rho}^*_\beta, \mathbf{r}^*_\beta) \leq L(\mathbf{x}) + \beta(\mathbf{x}, \boldsymbol{\lambda}, \boldsymbol{\rho}, \mathbf{r})$, which implies that

$$
L(\mathbf{x}^*_\beta) - L(\mathbf{x}) \leq \beta(p(\mathbf{x}, \boldsymbol{\lambda}, \boldsymbol{\rho}, \mathbf{r}) - \epsilon_\beta) \leq 0.
$$

Here, we prove that $(\mathbf{x}^*_\beta, \boldsymbol{\lambda}^*_\beta, \boldsymbol{\rho}^*_\beta, \mathbf{r}^*_\beta)$ is a global solution for 12 with $\epsilon = \epsilon_\beta$. $\qquad \square$

### A.3 PROOF OF THEOREM 2.9

*Proof.* Since $(\mathbf{x}^*_\epsilon, \boldsymbol{\lambda}^*_\epsilon, \boldsymbol{\rho}^*_\epsilon, \mathbf{r}^*_\epsilon)$ is an $\epsilon$-optimal solution of (10), we have

$$
L(\mathbf{x}^*_\epsilon) + \beta p(\mathbf{x}^*_\epsilon, \boldsymbol{\lambda}^*_\epsilon, \boldsymbol{\rho}^*_\epsilon, \mathbf{r}^*_\epsilon) \leq L(\mathbf{x}) + p(\mathbf{x}, \boldsymbol{\lambda}, \boldsymbol{\rho}, \mathbf{r}) + \epsilon.
\tag{23}
$$

Note that the conclusion in Proposition 2.8 still holds. Substituting $(\mathbf{x}, \boldsymbol{\lambda}, \boldsymbol{\rho}, \mathbf{r}) = (\mathbf{x}^*, \boldsymbol{\lambda}^*, \boldsymbol{\rho}^*, \mathbf{r}^*)$ with the fact $p(\mathbf{x}^*, \boldsymbol{\lambda}^*, \boldsymbol{\rho}^*, \mathbf{r}^*) = 0$, we have

$$
L(\mathbf{x}^*_\epsilon) + \beta p(\mathbf{x}^*_\epsilon, \boldsymbol{\lambda}^*_\epsilon, \boldsymbol{\rho}^*_\epsilon, \mathbf{r}^*_\epsilon) \leq L(\mathbf{x}^*, \boldsymbol{\lambda}^*, \boldsymbol{\rho}^*, \mathbf{r}^*) + \epsilon \leq L(\mathbf{x}^*_\epsilon) + \beta^* p(\mathbf{x}^*_\epsilon, \boldsymbol{\lambda}^*_\epsilon, \boldsymbol{\rho}^*_\epsilon, \mathbf{r}^*_\epsilon) + 2\epsilon,
$$

where the last inequality follows from (21). Then we have

$$
p(\mathbf{x}^*_\epsilon, \boldsymbol{\lambda}^*_\epsilon, \boldsymbol{\rho}^*_\epsilon, \mathbf{r}^*_\epsilon) \leq \frac{2\epsilon}{\beta - \beta^*}.
$$

Meawhile, $(\mathbf{x}^*_\epsilon, \boldsymbol{\lambda}^*_\epsilon, \boldsymbol{\rho}^*_\epsilon, \mathbf{r}^*_\epsilon)$ is feasible for the following problem

$$
\begin{aligned}
\min_{\mathbf{x}, \boldsymbol{\lambda}, \boldsymbol{\rho}, \mathbf{r}} \quad & L(\mathbf{x}) \\
\text{s.t.} \quad & p(\mathbf{x}, \boldsymbol{\lambda}, \boldsymbol{\rho}, \mathbf{r}) \leq p(\mathbf{x}^*_\epsilon, \boldsymbol{\lambda}^*_\epsilon, \boldsymbol{\rho}^*_\epsilon, \mathbf{r}^*_\epsilon).
\end{aligned}
\tag{24}
$$

From (23), we have $L(\mathbf{x}^*_\epsilon) - L(\mathbf{x}^*) \leq \beta(p(\mathbf{x}^*, \boldsymbol{\lambda}^*, \boldsymbol{\rho}^*, \mathbf{r}^*) - p(\mathbf{x}^*_\epsilon, \boldsymbol{\lambda}^*_\epsilon, \boldsymbol{\rho}^*_\epsilon, \mathbf{r}^*_\epsilon)) + \epsilon$. While $p(\mathbf{x}^*, \boldsymbol{\lambda}^*, \boldsymbol{\rho}^*, \mathbf{r}^*) = 0 \leq p(\mathbf{x}^*_\epsilon, \boldsymbol{\lambda}^*_\epsilon, \boldsymbol{\rho}^*_\epsilon, \mathbf{r}^*_\epsilon)$, we have $L(\mathbf{x}^*_\epsilon) - L(\mathbf{x}^*) \leq \epsilon$. $\qquad \square$

## B PROOF IN SECTION 3

In this section, we provide the proofs for the convergence results of our proposed algorithms in Section 3.

### B.1 PROOF OF LEMMA 3.5

*Proof.* From the definition $g_1(\mathbf{z}) = \mathcal{I}_{\mathcal{R}_1 \cap \mathcal{R}_1^*}(\mathbf{z})$, it holds that $\text{prox}_{t g_1} = \text{proj}_{\mathcal{R}_1 \cap \mathcal{R}_1^*}$ for $t > 0$. We define $P_L(\mathbf{z}) = L(\mathbf{x}) + \bar{\beta} p(\mathbf{z})$, then the update of $\mathbf{z}$ can be written as

$$
\mathbf{z}^{k+1} = \text{prox}_{\bar{\eta} g_1}(\mathbf{z}^k - \bar{\eta} \nabla P_L(\mathbf{z}^k)).
$$

From the $l_p$-smooth of $P_L(\mathbf{z})$, we have

$$
P_L(\mathbf{z}^{k+1}) \leq P_L(\mathbf{z}^k) + \langle \nabla P_L(\mathbf{z}^k), \mathbf{z}^{k+1} - \mathbf{z}^k \rangle + \frac{l_p}{2} \|\mathbf{z}^{k+1} - \mathbf{z}^k\|^2.
\tag{25}
$$

In addition, we denote $\bar{\mathbf{z}}^{k+1} = \mathbf{z}^k - \bar{\eta} \nabla_{\mathbf{z}} P_L(\mathbf{z}^k)$, then we have

$$
\langle \bar{\mathbf{z}}^{k+1} - \mathbf{z}^k, \mathbf{z}^{k+1} - \mathbf{z}^k \rangle \overset{(a)}{\leq} \bar{\eta} g_1(\mathbf{z}^k) - \bar{\eta} g_1(\mathbf{z}^{k+1}) \overset{(b)}{=} 0.
$$

where $(a)$ is from Theorem 6.39 in Beck (2017) and $(b)$ follows from the fact that $\mathbf{z}^{k+1}, \mathbf{z}^k \in \mathcal{R}_1 \cap \mathcal{R}_1^*$. Substituting the $\bar{\mathbf{z}}^{k+1} = \mathbf{z}^k - \bar{\eta} \nabla_{\mathbf{z}} P_L(\mathbf{z}^k)$, we have

$$\langle \nabla P_L(\mathbf{z}^k), \mathbf{z}^{k+1} - \mathbf{z}^k \rangle \leq -\frac{1}{\bar{\eta}} \|\mathbf{z}^{k+1} - \mathbf{z}^k\|^2. \tag{26}$$

Combining (25) and (26), we obtain that

$$P_L(\mathbf{z}^{k+1}) \leq P_L(\mathbf{z}^k) + (-\frac{1}{\bar{\eta}} + \frac{l_p}{2}) \|\mathbf{z}^{k+1} - \mathbf{z}^k\|^2,$$

which implies that $P_L(\mathbf{z}^{k+1}) - P_L(\mathbf{z}^k) \leq 0$ from $\bar{\eta} \leq \frac{l_p}{2}$. Utilizing the definition of $P(\mathbf{z})$, we have $L(\mathbf{x}^{k+1}) + \bar{\beta} p(\mathbf{z}^{k+1}) - L(\mathbf{x}^k) - \bar{\beta} p(\mathbf{z}^k) \leq 0$. In addition, we observe that $\{P_L(\mathbf{z}^k)\}$ is nonincreasing and bounded below, it converges. Therefore, $P_L(\mathbf{z}^k) - P_L(\mathbf{z}^{k+1}) \to 0$ as $k \to \infty$, along with $\|\mathbf{z}^{k+1} - \mathbf{z}^k\|^2 \to 0$ because $\|\mathbf{z}^{k+1} - \mathbf{z}^k\|^2 \leq 1/(\frac{1}{\bar{\eta}} - \frac{l_p}{2})(P_L(\mathbf{z}^k) - P_L(\mathbf{z}^{k+1}))$. Then we complete the proof.

$\square$

### B.2 PROOF OF THEOREM 3.6

*Proof.* According to the definition of $P_L(\mathbf{z})$ and $g_1(\mathbf{z})$, we know that (14) can be equivalently presented as the following form:

$$\min_{\mathbf{z}} \quad P_L(\mathbf{z}) + g_1(\mathbf{z}). \tag{27}$$

Then we define $M(\mathbf{z}) = \frac{1}{\bar{\eta}}[\mathbf{z} - \mathrm{prox}_{\bar{\eta}g_1}(\mathbf{z} - \bar{\eta}\nabla P_L(\mathbf{z}))] = \frac{1}{\bar{\eta}}[\mathbf{z} - \mathrm{proj}_{\mathcal{R}_1 \cap \mathcal{R}_1^*}(\mathbf{z} - \bar{\eta}\nabla P_L(\mathbf{z}))]$, representing the gradient mapping used for updating $\mathbf{z}$ in Algorithm 1 with respesct to (27). Then it holds that $M(\mathbf{z})$ is $(\frac{2}{\bar{\eta}} + l_p)$-Lipschitz continuous (Lemma 10.10 in Beck (2017)). Let $\bar{\mathbf{z}}$ is a limit point of $\{\mathbf{z}^k\}$. Then there exists a subsequence $\{\mathbf{z}^{k_j}\}$ converging to $\bar{\mathbf{z}}$. For any $j \geq 0$, we have

$$\|M(\bar{\mathbf{z}})\| \leq \|M(\mathbf{z}^{k_j}) - M(\bar{\mathbf{z}})\| + \|M(\mathbf{z}^{k_j})\| \leq (\frac{2}{\bar{\eta}} + l_p)\|\mathbf{z}^{k_j} - \bar{\mathbf{z}}\| + \|M(\mathbf{z}^{k_j})\|.$$

Based on proof for Lemma 3.5, we know that $\|M(\mathbf{z}^{k_j})\| \to 0$ as $j \to \infty$. Therefore, we conclude that $\|M(\bar{\mathbf{z}})\| = 0$ with taking the limit of the above inequality. According to the definition of $M(\mathbf{z})$, we observe that

$$\bar{\mathbf{z}} - \bar{\eta}\nabla P_L(\bar{\mathbf{z}}) \in \bar{\eta}\partial g_1(\bar{\mathbf{z}}),$$

which implies $\nabla P_L(\bar{\mathbf{z}}) \in \partial g_1(\bar{\mathbf{z}})$. From the first-order optimality condition, we conclude that $\bar{\mathbf{z}}$ serves as a stationary point of (14). $\square$

### B.3 PROOF OF LEMMA 3.8

**Theorem B.1.** *(Theorem 3.10 in Bubeck et al. (2015)) Let $f$ be $\alpha$-strongly convex and $\beta$-smooth on $\mathcal{X}$. Then projected gradient descent with $\eta = \frac{1}{\beta}$ satisfies for $t \geq 0$,*

$$\|x_{t+1} - x^*\|^2 \leq \exp(-\frac{t\beta}{\alpha})\|x_1 - x^*\|^2$$

According to Lemma 3.7, we know that the $\mathbf{z}$-subproblem in Algorithm 2 is $(t - \alpha_d)$-strongly convex and $l_d$-smooth, where we denote $\alpha_d = \frac{\beta}{2} - (1 + \beta)\alpha_l - \gamma$ and $l_d = \gamma + t + l_1 + \beta(l_1 + \alpha_l + 1)$. Therefore, the complexity for finding an $\epsilon_k$-optimal solution of $\mathbf{z}$-subproblem with projected grdient descent is $\mathcal{O}(\frac{l_d}{t - \alpha_d} \log(\frac{1}{\epsilon_k}))$.

### B.4 PROOF OF THEOREM 3.11

*Proof.* From the update of $\mathbf{u}$-subproblem, we have

$$\mathcal{L}_\gamma(\mathbf{z}^k, \mathbf{u}^{k+1}, \boldsymbol{\mu}^k) \leq \mathcal{L}_\gamma(\mathbf{z}^k, \mathbf{u}^k, \boldsymbol{\mu}^k).$$

Similarly, we derive from the iteration form and strong convexity of $\mathbf{z}$-subproblem that

$$\mathcal{L}_\gamma(\mathbf{z}^{k+1}, \mathbf{u}^{k+1}, \boldsymbol{\mu}^k) - \mathcal{L}_\gamma(\mathbf{z}^k, \mathbf{u}^{k+1}, \boldsymbol{\mu}^k) \geq \frac{2t - \alpha_d}{2} \|\mathbf{z}^{k+1} - \mathbf{z}^k\|^2.$$

Furthermore, we obtain from the update of $\boldsymbol{\mu}$ that

$$\mathcal{L}_\gamma(\mathbf{z}^{k+1}, \mathbf{u}^{k+1}, \boldsymbol{\mu}^k) - \mathcal{L}_\gamma(\mathbf{z}^{k+1}, \mathbf{u}^{k+1}, \boldsymbol{\mu}^{k+1})$$
$$= \langle \boldsymbol{\mu}^{k+1} - \boldsymbol{\mu}^k, \mathbf{u}^{k+1} - \mathbf{z}^{k+1} \rangle$$
$$= -\frac{1}{\gamma} \|\boldsymbol{\mu}^{k+1} - \boldsymbol{\mu}^k\|^2.$$

In summary, we obtain that

$$\mathcal{L}_\gamma(\mathbf{z}^k, \mathbf{u}^k, \boldsymbol{\mu}^k) - \mathcal{L}_\gamma(\mathbf{z}^{k+1}, \mathbf{u}^{k+1}, \boldsymbol{\mu}^{k+1}) \geq \frac{2t - \alpha_d}{2} \|\mathbf{z}^{k+1} - \mathbf{z}^k\|^2 - \frac{1}{\gamma} \|\boldsymbol{\mu}^{k+1} - \boldsymbol{\mu}^k\|^2 \quad (28)$$

We use the extended formula for Clark generalized gradient of a sum of two functions. $\partial(f_1 + f_2)(x) \subset \partial f_1(x) + \partial f_2(x)$ if $f_1$ and $f_2$ are finite at $\mathbf{x}$ and $f_2$ is differentiable at $x$. The equality holds if $f_1$ is regular at $x$ (Theorem 2.9.8 in Clarke (1990)). Then we have

$$\begin{aligned} B_k &\triangleq \partial_{\mathbf{z}} \left\{ L(\mathbf{x}^{k+1}) + \beta p(\mathbf{z}^{k+1}) + \langle \boldsymbol{\mu}^k, \mathbf{z}^{k+1} \rangle + \frac{\gamma}{2} \|\mathbf{u}^{k+1} - \mathbf{z}^{k+1}\|^2 + \frac{t}{2} \|\mathbf{z}^{k+1} - \mathbf{z}^k\|^2 \right\} \\ &= \partial_{\mathbf{z}} \{ L(\mathbf{x}^{k+1}) + \beta p(\mathbf{z}^{k+1}) \} + (\boldsymbol{\mu}^k + \gamma(\mathbf{u}^{k+1} - \mathbf{z}^{k+1})) + t(\mathbf{z}^{k+1} - \mathbf{z}^k) \\ &= \partial_{\mathbf{z}} \{ L(\mathbf{x}^{k+1}) + \beta p(\mathbf{z}^{k+1}) \} + \boldsymbol{\mu}^{k+1} + t(\mathbf{z}^{k+1} - \mathbf{z}^k). \end{aligned}$$
$$(29)$$

From the $\epsilon_k$-optimality condition, we obtain that $\|B_k\| \leq \epsilon_k$. From the assumption the $L$ and $p$ is bounded below, we know that

$$\mathcal{L}_\gamma(\mathbf{z}^k, \mathbf{u}^k, \boldsymbol{\mu}^k) = L(\mathbf{x}^k) + \beta p(\mathbf{z}^k) + g_1(\mathbf{z}^k) + g_2(\mathbf{u}^k) + \frac{\gamma}{2} \|\mathbf{u}^k + \mathbf{z}^k + \boldsymbol{\mu}^k/\gamma\|^2 - \|\boldsymbol{\mu}^k\|^2/2\gamma > -\infty$$

Therefore, $\mathcal{L}_\gamma(\mathbf{z}^k, \mathbf{u}^k, \boldsymbol{\mu}^k)$ is lower bounded by some $\mathcal{L}_b$. Moreover, with Assumption 3.10 holding, we find that $\mathcal{L}_\gamma(\mathbf{z}^0, \mathbf{u}^0, \boldsymbol{\mu}^0) - \mathcal{L}_\gamma(\mathbf{z}^{K+1}, \mathbf{u}^{K+1}, \boldsymbol{\mu}^{K+1}) + \frac{2}{\gamma} \sum_{k=1}^{K+1} \|\boldsymbol{\mu}^{k+1} - \boldsymbol{\mu}^k\|^2 < \infty$ for all $K \in \mathbb{N}$. We compress (28) from $k = 1$ to $K + 1$ and obtain that

$$\begin{aligned} &\mathcal{L}_\gamma(\mathbf{z}^0, \mathbf{u}^0, \boldsymbol{\mu}^0) - \mathcal{L}_\gamma(\mathbf{z}^{K+1}, \mathbf{u}^{K+1}, \boldsymbol{\mu}^{K+1}) + \frac{2}{\gamma} \sum_{k=1}^{K+1} \|\boldsymbol{\mu}^{k+1} - \boldsymbol{\mu}^k\|^2 \\ &\geq \frac{2t - \alpha_d}{2} \sum_{k=1}^{K+1} \|\mathbf{z}^{k+1} - \mathbf{z}^k\|^2 + \frac{1}{\gamma} \sum_{k=1}^{K+1} \|\boldsymbol{\mu}^{k+1} - \boldsymbol{\mu}^k\|^2. \end{aligned}$$
$$(30)$$

We take the minimum operation from $K$ iterations in (30) and obtain

$$\min_{k \leq K} \left\{ \frac{2t - \alpha_d}{2} \|\mathbf{z}^{k+1} - \mathbf{z}^k\|^2 + \frac{1}{\gamma} \|\boldsymbol{\mu}^{k+1} - \boldsymbol{\mu}^k\|^2 \right\} \leq \frac{\mathcal{L}_\gamma(\mathbf{z}^0, \mathbf{u}^0, \boldsymbol{\mu}^0) - \mathcal{L}_b}{K + 1}$$

Therefore, we observe that algorithm 2 execute $\mathcal{O}(1/\epsilon^2)$ iterations to find $(\mathbf{z}^{k+1}, \mathbf{u}^{k+1}, \boldsymbol{\mu}^{k+1})$ such that

$$\|\mathbf{z}^{k+1} - \mathbf{z}^k\| \leq \epsilon, \ \|\boldsymbol{\mu}^{k+1} - \boldsymbol{\mu}^k\| \leq \epsilon.$$

From the update of $\boldsymbol{\mu}$, we further derive that

$$\|\mathbf{u}^{k+1} - \mathbf{z}^{k+1}\| \leq \mathcal{O}(\epsilon)$$

From Assumption 3.9, it holds that

$$\text{dist}(-\boldsymbol{\mu}^{k+1}, \partial_{\mathbf{z}} \{ L(\mathbf{x}^{k+1}) + \beta p(\mathbf{z}^{k+1}) \}) \leq \mathcal{O}(\epsilon).$$

(29) and Now we consider the optimity condition of $\mathbf{u}$, then we have

$$0 \in \partial g_2(\mathbf{u}^{k+1}) + \boldsymbol{\mu}^k + \gamma(\mathbf{z}^{k+1} - \mathbf{u}^k).$$

Thus, we have

$$\text{dist}(-\boldsymbol{\mu}^{k+1}, \partial g_2(\mathbf{u}^{k+1})) \leq \gamma \|\boldsymbol{\mu}^{k+1} - \boldsymbol{\mu}^k\| = \mathcal{O}(\epsilon).$$

Then we conclude that $(\mathbf{z}^{k+1}, \mathbf{u}^{k+1}, \boldsymbol{\mu}^{k+1})$ attains an $\epsilon$-KKT point of (17). The proof is adapted from Theorem 4.1 in Lin et al. (2022). $\qquad \square$

## B.5 Extension to the Cases with Multiple Regularization terms

We focus on the case (2) involving multiple regularization terms. For this scenario, (10) simplifies to the following formulation:

$$\min_{\mathbf{x},\boldsymbol{\lambda},\boldsymbol{\rho},\mathbf{r}} \quad L(\mathbf{x}) + \beta p(\mathbf{x},\boldsymbol{\lambda},\boldsymbol{\rho},\mathbf{r}).$$
$$\text{s.t.} \quad R_i(\mathbf{x}) \le r_i, \|\boldsymbol{\rho}_i\|_* \le \lambda_i, i = 1, 2, ..., r, \tag{31}$$

where $p(\mathbf{x},\boldsymbol{\lambda},\boldsymbol{\rho},\mathbf{r}) = l(\mathbf{x}) + l^*(-\sum_{i=1}^{r}\boldsymbol{\rho}_i) + \sum_{i=1}^{r}\lambda_i r_i$. We adopt the notations $\mathbf{z} = (\mathbf{x},\boldsymbol{\lambda},\boldsymbol{\rho},\mathbf{r})$ and $\mathcal{R}_i, \mathcal{R}_i^*, i = 1, 2, ..., r$ defined in (13). Similar to Section 3.2, we denote $\mathcal{R}^* \triangleq \cap_{i=1}^{r}\mathcal{R}_i^*$ and consider variable decomposition for $\cap_{i=1}^{r}\mathcal{R}_i$. We define $g_i(\mathbf{z}) \triangleq \mathcal{I}_{\mathcal{R}_i \cap \mathcal{R}^*}(\mathbf{z}), i = 1, 2, ..., r$. Under this conditions, (31) can be rewritten as the following equivalent form,

$$\min_{\mathbf{z}} \quad L(\mathbf{x}) + \beta p(\mathbf{z}) + \sum_{i=1}^{r} g_i(\mathbf{z}). \tag{32}$$

Then we introduce an auxiliary variable $\mathbf{u}_i$ as follows,

$$\min_{\mathbf{z}} \quad L(\mathbf{x}) + \beta p(\mathbf{z}) + g_1(\mathbf{z}) + \sum_{i=1}^{r-1} g_2(\mathbf{u}_i)$$
$$\text{s.t.} \quad \mathbf{z} = \mathbf{u}_i, i = 1, 2, ..., r - 1. \tag{33}$$

We denote the constraints of (33) as $\sum_{i=1}^{r-1} \mathbf{I}_i \mathbf{u}_i + \mathbf{z} = 0$, where $\mathbf{I}_i$ is row full-rank matrix. (33) is a multi-block linearly constrained problem and its augmented Lagrangian function can be expressed as

$$\mathcal{L}_\gamma(\mathbf{z},\mathbf{u},\boldsymbol{\mu}) = L(\mathbf{x}) + \beta p(\mathbf{x}) + \sum_{i=1}^{r} g_i(\mathbf{u}_i) + \langle\boldsymbol{\mu}, \sum_{i=1}^{r-1}\mathbf{I}_i\mathbf{u}_i + \mathbf{z}\rangle + \frac{\gamma}{2}\|\sum_{i=1}^{r-1}\mathbf{I}_i\mathbf{u}_i + \mathbf{z}\|^2$$

Now, we employ multi-block ADMM to minimize equation 33, which cyclically update $\mathbf{u}_i, \mathbf{z}, \boldsymbol{\mu}$ by solving the $\mathbf{u}_i$- and $\mathbf{z}$- subproblems and adopt a dual ascent step for $\boldsymbol{\mu}$. We summarize the iterations in Algorithm 3.

---

**Algorithm 3** ADMM Framework for Problem (33)

---

1: Initialize $\mathbf{z}^0, \mathbf{u}^0, \sigma^0, \gamma$ and $t$.
2: **for** $k = 0, 1, 2, ...$ **do**
3:    $\mathbf{z}^{k+1} = \arg\min_{\mathbf{z}} \left\{ L(\mathbf{x}) + \beta p(\mathbf{x}) + \langle\boldsymbol{\mu}^k, \mathbf{z}\rangle + \frac{\gamma}{2}\|\sum_{i=1}^{r}\mathbf{I}_i\mathbf{u}_i^{k+1} + \mathbf{z}\|^2 + \frac{t}{2}\|\mathbf{z} - \mathbf{z}^k\|^2 \right\}.$
4:    **for** $i = 1, 2, ..., r - 1$ **do**
5:       $\mathbf{u}_i^{k+1} = \arg\min_{\mathbf{u}_i} \left\{ g_i(\mathbf{u}_i) + \langle\boldsymbol{\mu}^k, \mathbf{I}_i\mathbf{u}_i\rangle + \frac{\gamma}{2}\|\sum_{j<i}\mathbf{I}_j\mathbf{u}_j^{k+1} + \mathbf{I}_i\mathbf{u}_i + \sum_{j>i}\mathbf{I}_j\mathbf{u}_j^k + \mathbf{z}^k\|^2 \right\}.$
6:    **end for**
7:    $\boldsymbol{\mu}^{k+1} = \boldsymbol{\mu}^k + \gamma(\sum_{i=1}^{r-1}\mathbf{I}_i\mathbf{u}_i^{k+1} + \mathbf{z}^{k+1}).$
8: **end for**

---

**Theorem B.2.** *Suppose that the sequence $\{(\mathbf{z}^k, \mathbf{u}_i^k, \boldsymbol{\mu}^k)\}$ is bounded and $L(\mathbf{x}) + \beta p(\mathbf{x})$ is bounded below with bounded $(\mathbf{z}, \mathbf{u})$. Then Algorithm 3 can find an $\epsilon$-approximation KKT point $(\mathbf{z}^{k+1}, \mathbf{u}_i^{k+1}, \boldsymbol{\mu}^{k+1})$ of (equation 33).*

From the update of $\mathbf{u}$-subproblem, we have

$$\mathcal{L}_\gamma(\mathbf{z}^k, \mathbf{u}_{j \le i}^{k+1}, \mathbf{u}_{j > i}^k, \boldsymbol{\mu}^k) \le \mathcal{L}_\gamma(\mathbf{z}^k, \mathbf{u}_{j < i}^{k+1}, \mathbf{u}_{j \ge i}^k, \boldsymbol{\mu}^k).$$

Summing over $i = 1, 2, ..., r$, we have

$$\mathcal{L}_\gamma(\mathbf{z}^k, \mathbf{u}^{k+1}, \boldsymbol{\mu}^k) \le \mathcal{L}_\gamma(\mathbf{z}^k, \mathbf{u}^k, \boldsymbol{\mu}^k).$$

Consequently, the proof of Theorem B.2 follows from the proof of Theorem 3.11 in Appendix B.4.

## C  CLOSE-FORM PROJECTIONS

We observe that the set $\mathcal{R}_i$ and $\mathcal{R}_i^*$ takes the form of a norm cone, which are epigraphs of the norm and conjugate norm. The corresponding projections are orthogonal projections onto epigraphs, which are explored in Beck (2017); Wang et al. (2016).

**Theorem C.1.** *(Theorem 6.36 in Beck (2017)) Let*

$$C = \text{epi}(g) = \{(\mathbf{x}, t) | g(\mathbf{x}) \leq t\},$$

*where $g$ is convex. Then*

$$\text{proj}_C((\mathbf{x}, s)) = \begin{cases} (\mathbf{x}, s), & g(\mathbf{x}) \leq s, \\ (\text{prox}_{\lambda^* g}, s + \lambda^*), & g(\mathbf{x}) > s, \end{cases}$$

*where $\lambda^*$ is any positive root of the function*

$$\psi(\lambda) = g(\text{prox}_{\lambda g}(\mathbf{x}) - \lambda - s.)$$

*In addition, $\psi$ is nonincreasing.*

Based on Theorem C.1, the projections onto the epigraphs of the $l_1$ and $l_2$ norm can be calculated as follows. Let $C_1 = \{(\mathbf{x}, t) | \|\mathbf{x}\|_1 \leq t\}$ and $C_2 = \{(\mathbf{x}, t) | \|\mathbf{x}\|_2 \leq t\}$. Then it holds that (Example 6.37 and 6.38 in Beck (2017)),

$$\text{proj}_{C_1}((\mathbf{x}, s)) = \begin{cases} (\mathbf{x}, s), & \|\mathbf{x}\|_1 \leq s, \\ (\mathcal{T}_{\lambda^*}(\mathbf{x}), s + \lambda^*), & \|\mathbf{x}\|_1 > s, \end{cases}$$

We denote the proximal of $l_1$-norm as $\mathcal{T}_\lambda = \text{prox}_{\lambda\|\cdot\|_1}$, which is formed as

$$\mathcal{T}_\lambda(y) = [|y| - \lambda]_+ \text{sgn}(y) = \begin{cases} y - \lambda, & y \geq \lambda \\ 0, & |y| < \lambda, \\ y + \lambda, & y \leq -\lambda. \end{cases}$$

$\lambda^*$ is any positive root of the nonincreasing function $\varphi(\lambda) = \|\mathcal{T}_\lambda(\mathbf{x})\|_1 - \lambda - s$.

$$\text{proj}_{C_2}((\mathbf{x}, s)) = \begin{cases} (\frac{\|\mathbf{x}\|_2 + s}{2\|\mathbf{x}\|_2}\mathbf{x}, \frac{\|\mathbf{x}\|_2 + s}{2}), & \|\mathbf{x}\|_2 \geq |s|, \\ (\mathbf{0}, 0), & s < \|\mathbf{x}\|_2 < -s, \\ (\mathbf{x}, s), & \|\mathbf{x}\|_2 \leq s. \end{cases}$$

## D  EXPERIMENTS

We consider hyperparameter optimization for elastic net, group lasso, and sparse group lasso. These three models only use a combination of regularization terms $\|\cdot\|_1, \|\cdot\|_2, \frac{1}{2}\|\cdot\|_2^2$, as the form equation 2 . The elastic net (Zou & Hastie (2003)) is a linear combination of the lasso and ridge penalties, the group lasso (Yuan & Lin (2006)) is an extension of the Lasso with penalty to prede-fined groups of coefficients, and the sparse group lasso (Simon et al. (2013)) combines the group lasso and lasso penalties, which are designed to encourage sparsity and grouping of predictors (Feng & Simon (2018)). We consider hyperparameter optimization for elastic net, group lasso, and sparse group lasso. To compare the performance of each method, we calculate validation and test error with obtained LL minimizers in each experiment. Our competitors are implemented using code from https://github.com/SUSTech-Optimization/VF-iDCA, https://github.com/libra-licoho/LDMMA, and https://github.com/SUSTech-Optimization/BiC-GAFFA. Note that in the experiments, besides our method, solvers are all needed to solve the subproblems. and we uniformly apply the CVXPY package to them with the open source solvers ECOS and SCS only. All experiments are run on a computer with Intel(R) Core(TM) i7-10710U CPU @ 1.10GHz 1.61 GHz and 16.00 GB memory.

In our experiments, the parameter settings for LDPM are relatively loose. Since we use an exact penalty function, good results can be obtained with small penalty coeficient $\beta$ 1 or 10. Additionally, for smooth problems, we use the APG algorithm for the sub-problems, so the choice of step size $\alpha$ is not very sensitive due to the accelerated convergence rate. It is worth emphasizing that our algorithm is completely first-order and does not rely on any solver.

### D.1 EXPERIMENTS ON SYNTHETIC DATA

#### D.1.1 ELASTIC NET

The synthetic data is simulated in a similar manner as Gao et al. (2022) We sample the feature vectors $\mathbf{a}_i \in \mathbb{R}^p$ from a normal distribution with mean 0 and covariance $\mathrm{cor}(a_{ij}, a_{ik}) = 0.5^{|j-k|}$. We obtain the response vector $\mathbf{b}$ according to $b_i = \boldsymbol{\beta}^\top \mathbf{a}_i + \sigma \epsilon_i$, where $\beta_i$ is randomly 0 or 1 and $\sum_{i=1}^p \beta_i = 15$; the noise $\epsilon$ is sampled from the standard normal distribution, and $\sigma$ is chosen such that the signal-to-noise ratio SNR $\stackrel{\triangle}{=} \|A\boldsymbol{\beta}\|/\|\mathbf{b} - A\boldsymbol{\beta}\|$ equals 2.

We implement the algorithms we compared with the same parameters according to Chen et al. (2024). In our experiment, we set $\beta = 1$, $\gamma = 10$, and $t = 1$. Besides, we set the same initial guess $\lambda = (0.01, 0.01)$ and $r = (0.1, 0.1)$ as LDMMA and VF-iDCA, as well as the stopping criteria

$$\max \left\{ \frac{\|\mathbf{z}^{k+1} - \mathbf{z}^k\|}{\sqrt{1 + \|\mathbf{z}^k\|^2}}, t^{k+1} \right\} < 0.1, \tag{34}$$

We conduct repeated experiments with 30 randomly generated synthetic data, and calculate the mean and variance. The numerical results on elastic net are reported in Table 4 and we also plot the performance curve of the algorithms under each experiment setting in Figure 2. Overall, our algorithm achieves the lowest test error and validation error is also among lowest , along with its significantly lowest time cost, especially in large-scale data cases. Traditional gradient-free methods (grid search, random search, and TPE) have expensive search time cost and poor performance on test dataset. Gradient-based method IGJO perform slightly better on accuracy and efficiency, but it converges very slowly as shown in the figure. Considering the two solver-based algorithm, i.e. VF-iDCA and LDMMA, their validation error keeps very low in all experiments but they both suffer overfitting, where the test error goes higher as the iteration increases.

#### D.1.2 SPARSE GROUP LASSO

The synthetic data is simulated according to Gao et al. (2022). Each dataset contains 100 training data, 100 validation data and 100 test data. The feature vector $\mathbf{a}_i \in \mathbb{R}^p$ is sampled from the standard normal distribution. The response vector $\boldsymbol{b}$ is calculated by $b_i = \boldsymbol{\beta}^\top \mathbf{a}_i + \sigma \epsilon_i$, where $\boldsymbol{\beta} = \left[ \boldsymbol{\beta}^{(1)}, \boldsymbol{\beta}^{(2)}, \boldsymbol{\beta}^{(3)} \right]$, $\boldsymbol{\beta}^{(i)} = (1, 2, 3, 4, 5, 0, \ldots, 0)$, for $i = 1, 2, 3$. $\boldsymbol{\epsilon}$ are generated from the standard normal distribution, and $\sigma$ is chosen such that the SNR is 2.

We implement the algorithms we compared with the same parameters according to Chen et al. (2024). In our experiment, we set $\beta = 1$, $\gamma = 10$, and $t = 1$. Besides, we set the same initial guess $\lambda = 0.05\mathbf{1}$ and $r = 0.5\mathbf{1}$ as LDMMA and VF-iDCA, and tol=0.1.

We conduct experiments with different data scales and report numerical results over 30 repetitions in Table 5 with Figure 1. For each experiment, the generated datasets consist of n training, n/3 validation, and 100 test samples. LDPM achieves both lowest test error and validation error and outperforms other algorithms in terms of time cost. As the scale of data increases, our method consistently finds the best hyperparameters and model solutions, which indicates the superiority of LDPM in large-scale hyperparameter optimization. In comparison, gradient-free methods become extremely unstable when facing dozens of hyperparameters, while IGJO converges slowly and requires huge amount of computation. Due to the size of the problem, solving each subproblem (constrained optimization) is extremely time-consuming for LDMMA and VF-iDCA, even though they only needs several iterations to find good solutions. BiC-GAFFA runs as fast as LDPM in the gradient step iterations, but still requires some time to obtain an initial feasible point by solver in the first iteration.

#### D.1.3 GROUP LASSO

The synthetic data is generated in the same way as sparse group lasso. We set $\beta = 1$, and $\eta = 0.001$, with initial guess $\lambda = 0.1\mathbf{1}$ and $r = 0.5\mathbf{1}$ and $tol = 0.05$ in LDPM. We implement the rest algorithms with a slight modification for the problem with the same parameter setting in sparse group lasso experiments.

Table 1: Group lasso problems on synthetic data, where $p$ and $M$ represent the number of covariates and covariate groups, respectively, and $n$ represent the data scale described above.

| Settings | Methods | Time(s) | Val. Err. | Test Err. | Settings | Time(s) | Val. Err. | Test Err. |
|---|---|---|---|---|---|---|---|---|
| $n = 300$ $p = 600$ $M = 30$ | Grid | $85.18 \pm 4.61$ | $45.33 \pm 6.79$ | $48.84 \pm 6.76$ | $n = 450$ $p = 900$ $M = 60$ | $100.91 \pm 7.80$ | $45.38 \pm 5.74$ | $48.19 \pm 6.69$ |
| | Random | $79.11 \pm 5.10$ | $37.92 \pm 5.13$ | $45.66 \pm 6.34$ | | $93.06 \pm 6.72$ | $45.18 \pm 7.41$ | $43.86 \pm 4.89$ |
| | IGJO | $99.01 \pm 9.41$ | $34.86 \pm 5.80$ | $45.87 \pm 4.67$ | | $94.22 \pm 7.79$ | $38.75 \pm 7.72$ | $43.99 \pm 5.30$ |
| | VF-iDCA | $9.70 \pm 2.30$ | $27.21 \pm 5.37$ | $32.95 \pm 7.16$ | | $21.14 \pm 6.22$ | $24.07 \pm 2.20$ | $36.15 \pm 6.01$ |
| | LDMMA | $27.02 \pm 2.52$ | $25.76 \pm 3.60$ | $34.74 \pm 4.34$ | | $38.80 \pm 4.59$ | $26.95 \pm 4.33$ | $33.69 \pm 6.17$ |
| | GAFFA | $3.56 \pm 0.11$ | $29.73 \pm 6.48$ | $31.22 \pm 5.88$ | | $10.88 \pm 0.63$ | $25.84 \pm 7.19$ | $29.74 \pm 6.43$ |
| | LDPM | $\mathbf{0.55 \pm 0.04}$ | $\mathbf{17.42 \pm 3.74}$ | $\mathbf{25.10 \pm 3.68}$ | | $\mathbf{0.91 \pm 0.03}$ | $\mathbf{19.20 \pm 5.11}$ | $\mathbf{22.28 \pm 4.28}$ |
| $n = 300$ $p = 900$ $M = 60$ | Grid | $107.95 \pm 10.36$ | $46.13 \pm 5.54$ | $46.21 \pm 7.94$ | $n = 600$ $p = 1200$ $M = 150$ | $128.77 \pm 9.68$ | $45.33 \pm 6.43$ | $47.32 \pm 7.24$ |
| | Random | $95.02 \pm 7.27$ | $43.66 \pm 6.31$ | $42.18 \pm 6.77$ | | $131.50 \pm 7.66$ | $48.79 \pm 7.66$ | $48.91 \pm 9.13$ |
| | IGJO | $122.64 \pm 9.96$ | $30.56 \pm 6.46$ | $47.36 \pm 5.76$ | | $152.10 \pm 15.19$ | $37.21 \pm 6.89$ | $42.30 \pm 7.59$ |
| | VF-iDCA | $9.12 \pm 0.07$ | $24.40 \pm 5.62$ | $30.25 \pm 4.03$ | | $67.71 \pm 9.53$ | $27.53 \pm 5.16$ | $35.61 \pm 6.98$ |
| | LDMMA | $38.13 \pm 3.41$ | $24.94 \pm 6.68$ | $30.12 \pm 4.85$ | | $47.11 \pm 5.86$ | $18.51 \pm 4.09$ | $27.58 \pm 4.19$ |
| | GAFFA | $5.17 \pm 0.17$ | $28.39 \pm 6.22$ | $29.95 \pm 5.23$ | | $34.88 \pm 9.98$ | $25.39 \pm 5.41$ | $26.81 \pm 5.39$ |
| | LDPM | $\mathbf{0.86 \pm 0.02}$ | $\mathbf{20.69 \pm 3.88}$ | $\mathbf{27.04 \pm 4.58}$ | | $\mathbf{1.83 \pm 0.02}$ | $\mathbf{19.18 \pm 5.03}$ | $\mathbf{25.35 \pm 6.27}$ |

We conduct experiments with different data scales and report numerical results over 30 repetitions in Table 6 with Figure 3. For each experiment, the generated datasets consist of n training, n/3 validation, and 100 test samples. As a comparison to the Sparse Group Lasso experiment, we simply use APG to solve our problem thanks to the single regularization term (see 1), making our algorithm faster. LDPM achieves both lowest test error and validation error and outperforms other algorithms in terms of time cost. Performance of the rest algorithms is similar to that in the previous Sparse Group Lasso experiments. Note that in the experiments, We observe that the solver-based algorithms like LDMMA and VF-iDCA sometimes unable to run because of the solvers failure facing large scale data.

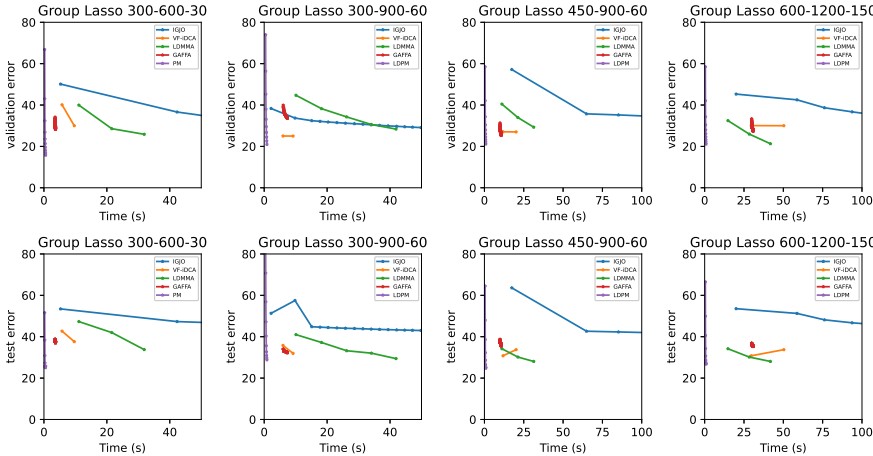

Figure 3: Comparison of the algorithms on Group Lasso problem for synthetic datasets in different scales

## D.2 EXPERIMENTS ON REAL DATA

We conduct experiments on the algorithm using real datasets from libsvm (Chang & Lin (2011)). The datasets we selected are gisette (Guyon et al. (2004)) and sensit (Duarte & Hu (2004)). Following the data participation rule as Gao et al. (2022), we randomly extracted 50, 25 examples as training set; 50, 25 examples as validation set, respectively; and the remaining for testing. We test different algorithms on the same partition for 30 repeated experiments. We perform hyperparameter tuning for elastic net and sparse group lasso on the two high-dimensional real datasets. The param-

eters are set the same as in the synthetic data experiments. We set The results are reported in Table 2, 3, and Figure 4, 5, showing the consistent effectiveness of our method.

Table 2: Elastic net problem on datasets gisette and sensit, where $|I_{tr}|$, $|I_{val}|$, $|I_{te}|$ and $p$ represent the number of training samples, validation samples, test samples and features, respectively.

| Settings | Methods | Time(s) | Val. Err. | Test Err. | Settings | Time(s) | Val. Err. | Test Err. |
|---|---|---|---|---|---|---|---|---|
| gisette $p = 5000$ $|I_{tr}| = 50$ $|I_{val}| = 50$ $|I_{te}| = 5900$ | Grid | $58.77 \pm 3.37$ | $0.25 \pm 0.04$ | $0.23 \pm 0.02$ | sensit $p = 78823$ $|I_{tr}| = 25$ $|I_{val}| = 25$ $|I_{te}| = 50$ | $1.08 \pm 0.15$ | $1.24 \pm 0.49$ | $1.22 \pm 0.47$ |
| | Random | $65.42 \pm 8.56$ | $0.25 \pm 0.04$ | $0.23 \pm 0.02$ | | $1.12 \pm 0.19$ | $1.21 \pm 0.58$ | $1.33 \pm 0.32$ |
| | TPE | $62.14 \pm 6.92$ | $0.24 \pm 0.03$ | $0.24 \pm 0.02$ | | $1.64 \pm 0.08$ | $1.19 \pm 0.55$ | $1.26 \pm 0.09$ |
| | IGJO | $18.10 \pm 2.77$ | $0.24 \pm 0.05$ | $0.22 \pm 0.03$ | | $0.47 \pm 0.12$ | $0.52 \pm 0.15$ | $0.71 \pm 0.19$ |
| | VF-iDCA | $12.85 \pm 2.25$ | $0.00 \pm 0.00$ | $0.19 \pm 0.01$ | | $0.76 \pm 0.17$ | $0.25 \pm 0.11$ | $0.55 \pm 0.06$ |
| | LDMMA | $10.99 \pm 0.87$ | $0.00 \pm 0.00$ | $0.20 \pm 0.01$ | | $0.20 \pm 0.05$ | $0.25 \pm 0.12$ | $0.51 \pm 0.09$ |
| | LDPM | $\mathbf{5.69 \pm 0.95}$ | $0.14 \pm 0.03$ | $\mathbf{0.18 \pm 0.01}$ | | $\mathbf{0.20 \pm 0.03}$ | $0.31 \pm 0.05$ | $\mathbf{0.49 \pm 0.05}$ |

Table 3: Sparse Group Lasso problem on datasets gisette and sensit, where $|I_{tr}|$, $|I_{val}|$, $|I_{te}|$ and $p$ represent the number of training samples, validation samples, test samples and features, respectively.

| Settings | Methods | Time(s) | Val. Err. | Test Err. | Settings | Time(s) | Val. Err. | Test Err. |
|---|---|---|---|---|---|---|---|---|
| gisette $p = 5000$ $|I_{tr}| = 50$ $|I_{val}| = 50$ $|I_{te}| = 5900$ $M = 10$ | Grid | $62.87 \pm 5.65$ | $0.34 \pm 0.05$ | $0.35 \pm 0.04$ | sensit $p = 78823$ $|I_{tr}| = 25$ $|I_{val}| = 25$ $|I_{te}| = 50$ $M = 10$ | $26.13 \pm 4.72$ | $1.39 \pm 0.32$ | $1.42 \pm 0.38$ |
| | Random | $63.25 \pm 6.10$ | $0.32 \pm 0.04$ | $0.33 \pm 0.03$ | | $29.38 \pm 4.92$ | $1.47 \pm 0.59$ | $1.37 \pm 0.49$ |
| | TPE | $60.28 \pm 9.43$ | $0.32 \pm 0.03$ | $0.31 \pm 0.04$ | | $38.60 \pm 6.59$ | $0.93 \pm 0.37$ | $1.03 \pm 0.45$ |
| | IGJO | $31.16 \pm 5.81$ | $0.28 \pm 0.03$ | $0.27 \pm 0.04$ | | $29.88 \pm 3.75$ | $0.97 \pm 0.38$ | $0.83 \pm 0.31$ |
| | VF-iDCA | $16.30 \pm 3.87$ | $0.10 \pm 0.02$ | $0.25 \pm 0.01$ | | $16.46 \pm 6.72$ | $0.43 \pm 0.19$ | $0.52 \pm 0.11$ |
| | LDMMA | $25.86 \pm 4.46$ | $0.30 \pm 0.03$ | $0.32 \pm 0.03$ | | $7.28 \pm 1.62$ | $0.47 \pm 0.10$ | $0.64 \pm 0.17$ |
| | GAFFA | $10.17 \pm 3.62$ | $0.26 \pm 0.03$ | $0.29 \pm 0.04$ | | $6.93 \pm 1.68$ | $0.60 \pm 0.21$ | $0.66 \pm 0.14$ |
| | LDPM | $\mathbf{7.35 \pm 0.84}$ | $0.20 \pm 0.03$ | $\mathbf{0.25 \pm 0.02}$ | | $\mathbf{3.72 \pm 1.61}$ | $0.45 \pm 0.11$ | $\mathbf{0.52 \pm 0.05}$ |

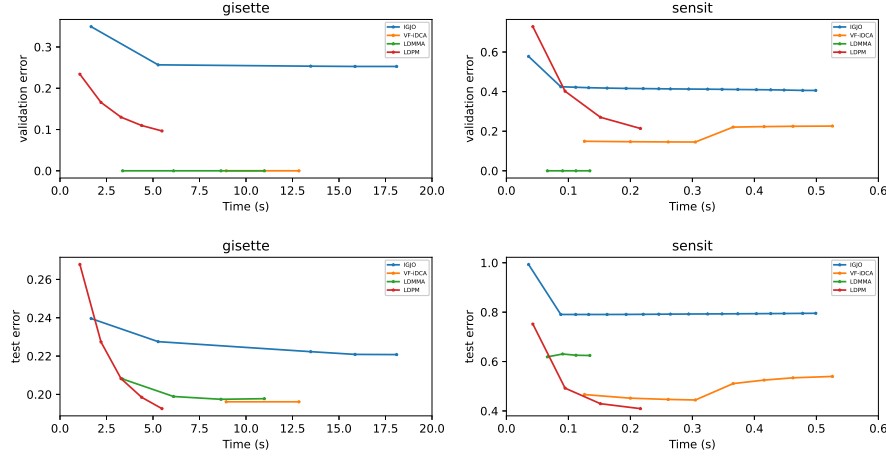

Figure 4: Comparison of the algorithms on Elastic Net problem for 2 datasets: gisette, sensit

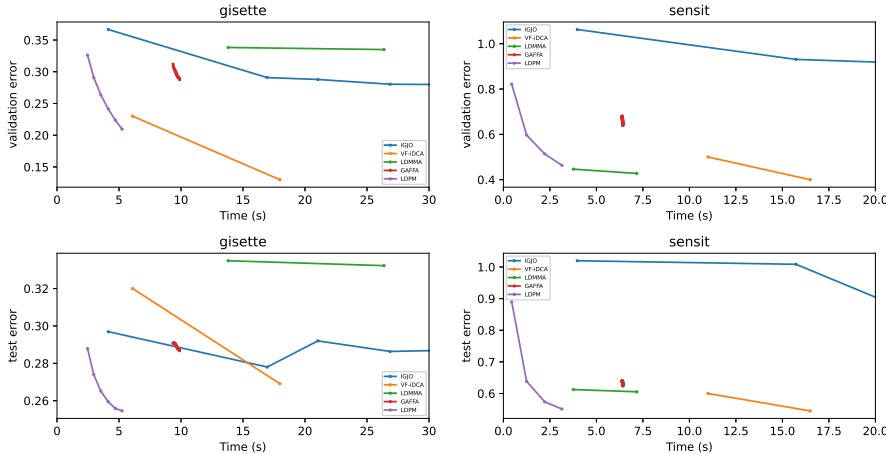

Figure 5: Comparison of the algorithms on Sparse Group Lasso problem for 2 datasets: gisette, sensit

Table 4: Elastic net problems on synthetic data, where $|I_{tr}|$, $|I_{val}|$, $|I_{te}|$ and $p$ represent the number of training observations, validation observations, predictors and features, respectively.

| Settings | Methods | Time(s) | Val. Err. | Test Err. | Settings | Time(s) | Val. Err. | Test Err. |
|---|---|---|---|---|---|---|---|---|
| | Grid | $5.76 \pm 0.33$ | $7.05 \pm 2.02$ | $6.98 \pm 1.14$ | | $11.72 \pm 1.32$ | $6.05 \pm 1.47$ | $6.49 \pm 0.82$ |
| $|I_{tr}| = 100$ | Random | $5.74 \pm 0.26$ | $7.01 \pm 2.01$ | $7.01 \pm 1.11$ | $|I_{tr}| = 100$ | $12.85 \pm 2.11$ | $6.04 \pm 1.45$ | $6.49 \pm 0.83$ |
| $|I_{val}| = 20$ | TPE | $6.55 \pm 0.26$ | $7.01 \pm 2.00$ | $7.01 \pm 1.09$ | $|I_{val}| = 100$ | $13.92 \pm 1.72$ | $6.03 \pm 1.44$ | $6.50 \pm 0.83$ |
| $|I_{te}| = 250$ | IGJO | $1.54 \pm 0.84$ | $4.99 \pm 1.69$ | $5.42 \pm 1.21$ | $|I_{te}| = 250$ | $3.37 \pm 1.85$ | $5.22 \pm 1.50$ | $5.72 \pm 0.91$ |
| $p = 250$ | VF-iDCA | $3.16 \pm 0.63$ | $2.72 \pm 1.57$ | $5.18 \pm 1.40$ | $p = 450$ | $6.08 \pm 2.24$ | $3.13 \pm 0.78$ | $5.39 \pm 0.92$ |
| | LDMMA | $1.64 \pm 0.07$ | $0.00 \pm 0.00$ | $6.97 \pm 0.79$ | | $3.95 \pm 0.22$ | $0.00 \pm 0.00$ | $6.56 \pm 0.70$ |
| | LDPM | $\mathbf{0.60 \pm 0.02}$ | $2.56 \pm 0.80$ | $\mathbf{4.92 \pm 0.51}$ | | $\mathbf{1.02 \pm 0.03}$ | $3.42 \pm 0.39$ | $\mathbf{4.23 \pm 0.37}$ |
| | Grid | $6.09 \pm 0.60$ | $6.39 \pm 1.09$ | $6.27 \pm 1.02$ | | $32.99 \pm 3.81$ | $7.81 \pm 1.53$ | $8.82 \pm 0.92$ |
| $|I_{tr}| = 100$ | Random | $6.44 \pm 1.28$ | $4.39 \pm 1.10$ | $6.27 \pm 1.05$ | $|I_{tr}| = 100$ | $33.82 \pm 2.66$ | $6.44 \pm 1.53$ | $8.67 \pm 0.94$ |
| $|I_{val}| = 100$ | TPE | $7.28 \pm 1.23$ | $6.37 \pm 1.09$ | $6.29 \pm 1.09$ | $|I_{val}| = 100$ | $42.70 \pm 3.89$ | $7.71 \pm 1.32$ | $8.43 \pm 0.80$ |
| $|I_{te}| = 250$ | IGJO | $3.86 \pm 2.09$ | $4.41 \pm 0.98$ | $4.31 \pm 0.95$ | $|I_{te}| = 100$ | $31.30 \pm 6.41$ | $7.78 \pm 1.12$ | $8.61 \pm 0.82$ |
| $p = 250$ | VF-iDCA | $4.74 \pm 1.77$ | $2.35 \pm 1.56$ | $4.47 \pm 1.11$ | $p = 2500$ | $23.21 \pm 4.96$ | $0.00 \pm 0.00$ | $4.61 \pm 0.77$ |
| | LDMMA | $0.98 \pm 0.09$ | $0.00 \pm 0.00$ | $5.61 \pm 0.77$ | | $16.26 \pm 1.44$ | $0.00 \pm 0.00$ | $5.67 \pm 1.21$ |
| | LDPM | $\mathbf{0.73 \pm 0.08}$ | $3.41 \pm 0.48$ | $\mathbf{3.51 \pm 0.40}$ | | $\mathbf{4.83 \pm 0.08}$ | $1.65 \pm 0.14$ | $\mathbf{4.37 \pm 0.65}$ |

Table 5: Sparse group lasso problems on synthetic data, where $p$ and $M$ represent the number of covariates and covariate groups, respectively, and $n$ represent the data scale described above.

| Settings | Methods | Time(s) | Val. Err. | Test Err. | Settings | Time(s) | Val. Err. | Test Err. |
|---|---|---|---|---|---|---|---|---|
| | Grid | $96.36 \pm 2.88$ | $44.73 \pm 5.29$ | $47.34 \pm 5.91$ | | $103.68 \pm 5.49$ | $44.68 \pm 4.31$ | $46.00 \pm 4.43$ |
| $n = 300$ | Random | $83.02 \pm 3.01$ | $35.17 \pm 5.95$ | $47.43 \pm 5.54$ | $n = 450$ | $108.64 \pm 8.84$ | $37.87 \pm 4.21$ | $46.25 \pm 5.52$ |
| $p = 600$ | IGJO | $117.58 \pm 7.28$ | $31.93 \pm 4.07$ | $46.36 \pm 3.72$ | $p = 900$ | $120.35 \pm 6.46$ | $30.56 \pm 4.02$ | $46.70 \pm 4.01$ |
| $M = 30$ | VF-iDCA | $19.00 \pm 0.55$ | $26.96 \pm 2.58$ | $36.84 \pm 5.33$ | $M = 60$ | $29.63 \pm 2.91$ | $26.38 \pm 3.40$ | $37.58 \pm 5.90$ |
| | LDMMA | $24.62 \pm 0.13$ | $22.70 \pm 2.03$ | $31.44 \pm 4.72$ | | $22.72 \pm 2.15$ | $23.93 \pm 2.32$ | $31.03 \pm 4.08$ |
| | GAFFA | $2.59 \pm 0.02$ | $27.42 \pm 3.28$ | $28.45 \pm 4.74$ | | $11.52 \pm 0.79$ | $22.21 \pm 3.03$ | $29.81 \pm 4.66$ |
| | LDPM | $\mathbf{1.26 \pm 0.03}$ | $\mathbf{15.11 \pm 1.62}$ | $\mathbf{23.48 \pm 2.40}$ | | $\mathbf{1.95 \pm 0.04}$ | $\mathbf{19.39 \pm 1.51}$ | $\mathbf{25.11 \pm 2.35}$ |
| | Grid | $104.23 \pm 4.05$ | $45.63 \pm 4.13$ | $44.86 \pm 5.09$ | | $117.09 \pm 6.34$ | $48.94 \pm 4.11$ | $49.41 \pm 7.62$ |
| $n = 300$ | Random | $98.17 \pm 6.85$ | $40.04 \pm 5.36$ | $46.77 \pm 6.70$ | $n = 600$ | $126.3 \pm 5.57$ | $49.41 \pm 6.55$ | $52.49 \pm 9.46$ |
| $p = 900$ | IGJO | $117.14 \pm 7.44$ | $31.59 \pm 4.97$ | $45.98 \pm 4.94$ | $p = 1200$ | $169.76 \pm 9.44$ | $39.75 \pm 5.14$ | $46.49 \pm 7.48$ |
| $M = 60$ | VF-iDCA | $44.31 \pm 1.45$ | $23.21 \pm 3.36$ | $31.92 \pm 3.54$ | $M = 150$ | $45.12 \pm 3.10$ | $23.66 \pm 4.53$ | $35.09 \pm 3.14$ |
| | LDMMA | $37.50 \pm 0.21$ | $26.23 \pm 3.47$ | $32.09 \pm 3.75$ | | $36.14 \pm 3.65$ | $18.61 \pm 2.32$ | $27.81 \pm 3.43$ |
| | GAFFA | $5.11 \pm 0.10$ | $26.83 \pm 3.53$ | $30.38 \pm 3.60$ | | $33.03 \pm 4.63$ | $24.34 \pm 4.19$ | $26.05 \pm 5.13$ |
| | LDPM | $\mathbf{1.87 \pm 0.05}$ | $\mathbf{19.32 \pm 2.62}$ | $\mathbf{27.14 \pm 2.79}$ | | $\mathbf{3.08 \pm 0.07}$ | $\mathbf{17.35 \pm 2.04}$ | $\mathbf{24.21 \pm 2.74}$ |

Table 6: Group lasso problems on synthetic data, where $p$ and $M$ represent the number of covariates and covariate groups, respectively, and $n$ represent the data scale described above.

| Settings | Methods | Time(s) | Val. Err. | Test Err. | Settings | Time(s) | Val. Err. | Test Err. |
|---|---|---|---|---|---|---|---|---|
| | Grid | $85.18 \pm 4.61$ | $45.33 \pm 6.79$ | $48.84 \pm 6.76$ | | $100.91 \pm 7.80$ | $45.38 \pm 5.74$ | $48.19 \pm 6.69$ |
| $n = 300$ | Random | $79.11 \pm 5.10$ | $37.92 \pm 5.13$ | $45.66 \pm 6.34$ | $n = 450$ | $93.06 \pm 6.72$ | $45.18 \pm 7.41$ | $43.86 \pm 4.89$ |
| $p = 600$ | IGJO | $99.01 \pm 9.41$ | $34.86 \pm 5.80$ | $45.87 \pm 4.67$ | $p = 900$ | $94.22 \pm 7.79$ | $38.75 \pm 7.72$ | $43.99 \pm 5.30$ |
| $M = 30$ | VF-iDCA | $9.70 \pm 2.30$ | $27.21 \pm 5.37$ | $32.95 \pm 7.16$ | $M = 60$ | $21.14 \pm 6.22$ | $24.07 \pm 2.20$ | $36.15 \pm 6.01$ |
| | LDMMA | $27.02 \pm 2.52$ | $25.76 \pm 3.60$ | $34.74 \pm 4.34$ | | $38.80 \pm 4.59$ | $26.95 \pm 4.33$ | $33.69 \pm 6.17$ |
| | GAFFA | $3.56 \pm 0.11$ | $29.73 \pm 6.48$ | $31.22 \pm 5.88$ | | $10.88 \pm 0.63$ | $25.84 \pm 7.19$ | $29.74 \pm 6.43$ |
| | LDPM | $\mathbf{0.55 \pm 0.04}$ | $\mathbf{17.42 \pm 3.74}$ | $\mathbf{25.10 \pm 3.68}$ | | $\mathbf{0.91 \pm 0.03}$ | $\mathbf{19.20 \pm 5.11}$ | $\mathbf{22.28 \pm 4.28}$ |
| | Grid | $107.95 \pm 10.36$ | $46.13 \pm 5.54$ | $46.21 \pm 7.94$ | | $128.77 \pm 9.68$ | $45.33 \pm 6.43$ | $47.32 \pm 7.24$ |
| $n = 300$ | Random | $95.02 \pm 7.27$ | $43.66 \pm 6.31$ | $42.18 \pm 6.77$ | $n = 600$ | $131.50 \pm 11.36$ | $48.79 \pm 7.66$ | $48.91 \pm 9.13$ |
| $p = 900$ | IGJO | $122.64 \pm 9.96$ | $30.56 \pm 6.46$ | $47.36 \pm 5.76$ | $p = 1200$ | $152.10 \pm 15.19$ | $37.21 \pm 6.89$ | $42.30 \pm 7.59$ |
| $M = 60$ | VF-iDCA | $9.12 \pm 0.07$ | $24.40 \pm 5.62$ | $30.25 \pm 4.03$ | $M = 150$ | $67.71 \pm 9.53$ | $27.53 \pm 5.16$ | $35.61 \pm 6.98$ |
| | LDMMA | $38.13 \pm 3.41$ | $24.94 \pm 6.68$ | $30.12 \pm 4.85$ | | $47.11 \pm 5.86$ | $18.51 \pm 4.09$ | $27.58 \pm 4.19$ |
| | GAFFA | $5.17 \pm 0.17$ | $28.39 \pm 6.22$ | $29.95 \pm 5.23$ | | $34.88 \pm 9.98$ | $25.39 \pm 5.41$ | $26.81 \pm 5.39$ |
| | LDPM | $\mathbf{0.86 \pm 0.02}$ | $\mathbf{20.69 \pm 3.88}$ | $\mathbf{27.04 \pm 4.58}$ | | $\mathbf{1.83 \pm 0.02}$ | $\mathbf{19.18 \pm 5.03}$ | $\mathbf{25.35 \pm 6.27}$ |