# OpenReview forum: "Lower-level Duality Based Penalty Methods for Hyperparameter Optimization"
_ICLR.cc/2025/Conference — ICLR 2025 Conference Withdrawn Submission_

### Official Review · Reviewer_kYC5 · 2024-10-25

**Soundness:** 2
**Presentation:** 3
**Contribution:** 2
**Rating:** 5
**Confidence:** 5

**Summary:**

This paper proposes a penalty-based framework to reformulate certain bilevel optimization problems into single-level optimization tasks. The approach is supported by equivalence and convergence guarantees, providing a new perspective on handling bilevel structures.

**Strengths:**

The work addresses a notable trend in bilevel optimization: converting bilevel problems into single-level formulations. The approach is particularly valuable for the challenging case where the lower-level problem lacks smoothness.

**Weaknesses:**

While the methodology is intriguing, the added value of this contribution is not entirely clear. For instance, consider the case of a single hyperparameter $\lambda$ in a bilevel problem, such as (2), where cross-validation is required. The proposed reformulation introduces multiple variables encapsulated in $r$ as well as an additional hyperparameter $\beta$, which also requires cross-validation. This merely shifts the bilevel challenge from $\lambda$ to $\beta$. Moreover, for problems like sparse group lasso and elastic net, understanding the impact of $\lambda$ on $\lambda \mapsto x_\lambda$ is arguably more intuitive than analyzing $\beta$ on $r_\beta$.

The experimental section lacks motivation regarding the selection of baseline algorithms, and more detailed analysis would enhance clarity. For instance, reporting hyperparameter estimates and solution support would help. Additionally, the figures need improvement for readability.

Finally, certain imprecisions should be addressed to prevent misconceptions for unfamiliar readers (see examples in “Questions”).

**Questions:**

- Prior work, such as [1,2], tackles bilevel problems with non-smooth lower-levels by introducing smooth lower-level algorithms. How does your method compare to these?
- There are sign inconsistencies from (4) to (5), and it seems incorrect to move the max over $\rho$ (which becomes min) from the inequality in (6) and merge it with the min over $x$ and $\lambda$. It is then correct once the penalized objective is introduced. Do you concur?
- Before Assumption 2.1, it is stated that “the validity of (6) depends on the following assumption.” This is debatable, as (6) should hold even without an explicit closed form. Could you clarify?

[1] P. Ochs, R. Ranftl, T. Brox, and T. Pock. Techniques for gradient-based bilevel optimization with non-smooth lower level problems. Journal of Mathematical Imaging and Vision. 2016

[2] J. Frecon, S. Salzo, and M. Pontil. Bilevel learning of the group lasso structure. NeurIPS 2018.

---

### Official Review · Reviewer_ppPZ · 2024-11-01

**Soundness:** 3
**Presentation:** 3
**Contribution:** 1
**Rating:** 3
**Confidence:** 4

**Summary:**

In this work the authors present two algorithms for solving bilevel optimization problems used to model the problem of hyperparameter estimation in several learning applications. The approach is based on the reformulation of the bilevel problem as a penalised problem, which is done by exploiting strong duality and conjugation techniques. The structure of the penalised problem is then used to design a proximal gradient algorithm (since the resulting constraints are prox-explicit) and an ADMM-type algorithm. Several numerical results are showed.

**Strengths:**

The reformulation of the bilevel problem into a penalised one whose solutions are very close to the ones of the original one is very interesting a surely the main point of the whole paper. Numerical validations are convincing.

**Weaknesses:**

The key idea described above (which is good) has been presented alredy in at least a large paper written presumably by the same authors and with a significant amount of overlapping content with respect to this one.
* He Chen, Haochen Xu, Rujun Jiang, Anthony Man-Cho So, "Lower-level Duality Based Reformulation and Majorization Minimization Algorithm for Hyperparameter Optimization", Proceedings of The 27th International Conference on Artificial Intelligence and Statistics, PMLR 238:784-792, 2024.

I checked carefully and despite the attempts of the authors to slightly change notations and, in some respect, the whole paper presentation, some paragraphs are almost repeated verbatim. This is the case for instance of the main and most interesting part of the papers covering the reformulation of the bilevel optimisation problem, which, as mentioned above, are important and interesting contribution, but, unfortunately, covered and published elsewhere.
The rest of the paper is just the presentation of two algorithms different from the MM (LDMMA) one presented in the paper above. Similar numerical tests are performed (on elastic net and group lasso). The one on Group Lasso shows significant numerical improvements, while the other one shows only better test errors in comparisons with LDMMA which enjoys however better validation errors.

**Questions:**

I am not sure that given the concern above there is a chance I will consider changing my rating. But the least the authors could try to do is putting their work into the larger context of why these algorithms are needed, for instance when the MM one presented in their previous work does not apply or pefrorms very poorly.

**Details Of Ethics Concerns:**

see above

---

### Official Review · Reviewer_sRuq · 2024-11-03

**Soundness:** 3
**Presentation:** 2
**Contribution:** 3
**Rating:** 6
**Confidence:** 3

**Summary:**

This paper addresses hyperparameter optimization (HO) in the context of nonsmooth regularizers by proposing a novel penalty method based on lower-level duality (LDPM), which avoids any implicit value functions and high-complexity subproblems. Under certain conditions, the penalized problem is illustrated to be closely approximates the optimal solutions of the original HO. The authors introduce two fully first-order algorithms to solve the penalized problems and provide theoretical proof of their convergence. Numerical experiments demonstrate the efficiency and superiority of LDPM.

**Strengths:**

1.The authors provide a novel penalty method based on lower-level duality, avoiding any implicit value functions and high-complexity subproblems.

2.Two fully first-order algorithms based on proximal techniques and the alternating direction method of multipliers are proposed with theoretical proof of convergence and promising numerical experiments.

3.The proposed algorithms does not rely on any open-source libraries or commercial optimization solvers.

**Weaknesses:**

1.The strong convexity of $l(x)$ may hinder LDPM’s application to more general problems.

2.The convergence results for the cases of multiple regularization terms, as detailed in Appendix B.5-Theorem B.2, do not specify the requisite number of iterations as outlined in Theorem 3.11.

**Questions:**

1.In equation (5), why could the max operator be dropped to obtain the reformulation (6) for (2)? Is it an equivalent reformulation, or just done by intuition? In fact, by Fenchel’s inequality, the inequality constraint of (6) becomes equality constraint. Explain it more clearly.

2.The constraint of (6) is equivalent to (7) when $\|\rho_i\|_*\le\lambda_i$, so why exclude the case of $\|\rho_i\|_*>\lambda_i$? Explain it more specifically.

Minor:
1.In equation (4) and (5), $\mathcal{R}^*_i\left(-\frac{\rho_i}{\lambda_i}\right)$ should be $\mathcal{R}^*_i\left(\frac{\rho_i}{\lambda_i}\right)$.

2.In line 184, 2 should be (2). Similar typos arise in line 770.

3.In the proof of Lemma 2.6, RHS of the (b) inequality: the sign “-” before “$\min_{\rho}$” should be “+”; RHS of the (c) equality: last “+” should be “-”.

---

### Official Review · Reviewer_QacY · 2024-11-04

**Soundness:** 2
**Presentation:** 2
**Contribution:** 1
**Rating:** 3
**Confidence:** 5

**Summary:**

The paper tackles the problem of solving bi-level optimization problems with composite non-smooth inner optimization problems, more specifically with inner problems of the form $\min_x l(x) + \sum_i^r \lamdba_i R_i(x)$ with $R_i$ defines as norms.

**Strengths:**

The paper is easy to follow, the tackled problem is hard, and the idea is interesting.

**Weaknesses:**

- Lack of literature review: how does the paper compare to [1]?
- Assumption 3.1 is **very very** strong, and to the best of my knowledge is not true for the $\ell_1 + \ell_2$- norm for instance.
- Following the later question, how do you compute it in the sparse-group Lasso experiments?
- Algorithm 2 has **a lot** of hyperparameters, how do you select
$\lambda^0, \rho^0, r^0, \beta, \gamma, t$?

Experiments:
- what is the number of inner optimization steps to solve the problems in lines 3 and 4 in Algorithm 2?
- overall, in my experience, it is hard to gain insight from this kind of experiment: each method is so sensitive to hyperparameter tuning. Except huge grid-search to select the hyperparameters, I do not see how to properly compare all the methods (this is a recurrent problem in bilevel optimization)
- overall the provided experiments are very limited, experiments on real data are not provided in the main text. The data are not even described in the main text: this brings confusion to the reader.
- In addition, it seems to me the real data has not been used correctly: in Appendix D. 2 it is written ¨The datasets we selected are Gisette (Guyon et al. (2004)) and sensit (Duarte & Hu (2004)). Following the data participation rule as Gao et al. (2022), we randomly extracted 50, 25 examples as training set 50, 25 examples as validation set, respectively; and the remaining for testing¨, what does this mean exactly? Gisette on libsvm has a dedicated separated test set.



[1] Mehmood, Sheheryar, and Peter Ochs. "Differentiating the value function by using convex duality." International Conference on Artificial Intelligence and Statistics

**Questions:**

see weaknesses

---

### Author Response · Authors · 2024-11-24

We are grateful for the valuable feedback provided by the reviewers. After careful consideration, we have decided to withdraw the paper in order to make necessary refinements and improvements. We will address the reviewers' suggestions and enhance the quality of the work for future submissions.

---

### Note · Authors · 2024-11-24

I have read and agree with the venue's withdrawal policy on behalf of myself and my co-authors.